# Immunological synapse formation between T regulatory cells and cancer-associated fibroblasts promotes tumour development

Athina Varveri[1,2], Miranta Papadopoulou[1,2], Zacharias Papadovasilakis[2,3], Ewoud B. Compeer [4], Aigli-Ioanna Legaki[1], Anastasios Delis[5], Vasileia Damaskou[6], Louis Boon[7], Sevasti Papadogiorgaki[8], Martina Samiotaki[9], Periklis G. Foukas[6], Aristides G. Eliopoulos [10], Aikaterini Hatzioannou[10,11], Themis Alissafi [4,10], Michael L. Dustin [4] & Panayotis Verginis [1,2,3,11] ✉

Cancer-associated fibroblasts (CAFs) have emerged as a dominant non-hematopoietic cell population in the tumour microenvironment, serving diverse functions in tumour progression. However, the mechanisms via which CAFs influence the anti-tumour immunity remain poorly understood. Here, using multiple tumour models and biopsies from cancer patients, we report that α-SMA⁺ CAFs can form immunological synapses with Foxp3⁺ regulatory T cells (Tregs) in tumours. Notably, α-SMA⁺ CAFs can phagocytose and process tumour antigens and exhibit a tolerogenic phenotype which instructs movement arrest, activation and proliferation in Tregs in an antigen-specific manner. Moreover, α-SMA⁺ CAFs display double-membrane structures resembling autophagosomes in their cytoplasm. Single-cell transcriptomic data showed an enrichment in autophagy and antigen processing/presentation pathways in α-SMA-expressing CAF clusters. Conditional knockout of *Atg5* in α-SMA⁺ CAFs promoted inflammatory re-programming in CAFs, reduced Treg cell infiltration and attenuated tumour development. Overall, our findings reveal an immunosuppressive mechanism entailing the formation of synapses between α-SMA⁺ CAFs and Tregs in an autophagy-dependent manner.

Cancer immunotherapy has revolutionized the field of oncology, showing remarkable efficacy in prolonging the survival of patients with rapidly growing cancers. Immune-based treatments are currently used as first-line therapies for many cancer indications, yet, induction of durable and sustained responses remains limited in a significant portion of patients[1]. A comprehensive understanding of the mechanisms underlying the limited beneficial effects of immunotherapy is of urgent need and will facilitate the identification of predictive biomarkers for

[1]Center of Clinical, Experimental Surgery & Translational Research, Biomedical Research Foundation Academy of Athens, Athens, Greece. [2]Laboratory of Immune Regulation and Tolerance, Division of Basic Sciences, Medical School, University of Crete, Heraklion, Greece. [3]Institute of Molecular Biology and Biotechnology, Foundation for Research and Technology, Heraklion, Greece. [4]Kennedy Institute of Rheumatology, Nuffield Department of Orthopaedics, Rheumatology and Musculoskeletal Sciences, University of Oxford, Oxford, UK. [5]Center of Basic Research, Biomedical Research Foundation Academy of Athens, Athens, Greece. [6]2nd Department of Pathology, National and Kapodistrian University of Athens, Attikon University Hospital, Athens, Greece. [7]JJP Biologics, Warsaw, Poland. [8]Electron Microscopy Laboratory, University of Crete, Heraklion, Greece. [9]Institute for Bioinnovation, Biomedical Sciences Research Centre Alexander Fleming, Vari Athens 166 72, Greece. [10]Laboratory of Biology, School of Medicine, Medical School National and Kapodistrian University of Athens, Athens, Greece. [11]Institute for Clinical Chemistry and Laboratory Medicine, University Hospital and Faculty of Medicine Carl Gustav Carus of TU Dresden, Dresden, Germany. ✉e-mail: pverginis@uoc.gr

patient stratification and personalized therapy. Resistance to cancer immunotherapy is influenced by tumour cell-intrinsic factors, such as the mutation burden, defects concerning the antigen presentation machinery, and high expression of inhibitory immune checkpoint molecules (i.e., PD-L1)[2], as well as extrinsic mechanisms which entail the recruitment of immunosuppressive cells, such as regulatory T cells (Treg cells) and myeloid-derived suppressor cells (MDSCs), into the tumour microenvironment (TME)[2,3]. It is established today that the TME ecosystem is highly heterogeneous, and although the role of hematopoietic cells in tumour development and immunotherapy efficacy is extensively studied, the mechanisms governing non-hematopoietic cell function and contribution to therapy resistance are largely ignored.

Cancer-associated fibroblasts (CAFs) constitute the most abundant non-hematopoietic stromal cell population in the TME of diverse solid tumour types[4,5]. Increased frequencies of CAFs are associated with worse prognosis in cancer patients[6–8], while their presence is linked to almost all hallmarks of cancer[9–11]. To this end, CAFs promote the stiffness of the TME via the production and secretion of collagens, fibrillar proteins and matrix metalloproteinases (MMPs)[5] and thus are key drivers of the desmoplastic reaction occurring in tumours. Also, CAFs influence the characteristics of the TME by producing and remodelling the extracellular matrix (ECM), as well as by secreting a plethora of chemokines, cytokines and growth factors[12,13]. As a result, CAFs are considered as promoters of aggressive tumour cell behaviours, such as epithelial to mesenchymal transition, invasiveness, metastasis and resistance to therapy[5,13]. Despite this evidence, more recently, CAFs have also been reported to restrain tumour progression[4,14,15], suggesting the existence of several CAF subtypes in the TME, which may regulate different aspects of cancer biology and thus exert pro- or anti-tumorigenic roles. Indeed, single-cell RNA sequencing (scRNA-seq) technology has shed light on the concept of CAF heterogeneity, which has been addressed in several cancer types, including pancreatic ductal adenocarcinoma (PDAC), breast cancer and melanoma[16–18]. Thus, mutually exclusive CAF sub-populations coexist in the TME, characterized by diverse expression levels of proteins, such as alpha-smooth muscle actin (α-SMA), fibroblast-activation protein α (FAPα), fibroblast specific protein-1 (FSP-1, also called S100A4), neuron-glial antigen-2 (NG2) and platelet-derived growth factor receptor-β (PDGFR-β)[13,19]. Despite their extensive characterization, the molecular mechanisms via which CAFs modulate anti-tumour immunity remain obscure. Towards this, CAFs are proposed to contribute to the formation of a tolerogenic TME, by influencing the density and composition of matrix components[4,13]. Furthermore, through the secretion of immunosuppressive mediators (e.g., CXCL12/SDF1, M-CSF, IL-6 and CCL2/MCP-1), CAFs contribute to the recruitment and differentiation of myeloid cells in the TME[20]. In addition, CAF secretome is shown to promote T cell anergy or apoptosis and to attenuate the activation and effector function of natural killer cells[20]. Finally, while under certain conditions CAFs were presented to interact with Treg cells, impeding T cell responses, other studies provided evidence for a stimulatory role of CAF subsets through promotion of CD4+ T cell proliferation[20–23]. Overall, it seems plausible that CAFs may be instrumental in directing T cell responses in the TME, however evidence supporting such hypotheses remains limited.

In this work, we describe the crucial role of α-SMA+ CAFs in forming synapses with Treg cells which facilitate tumour development. Treg cells were found in close proximity with α-SMA+ CAFs in diverse types of experimental tumour models and tumour biopsies from patients with melanoma and colorectal cancer. Time-lapse microscopy revealed a movement arrest of Treg cells upon exposure to CAFs in an antigen-specific manner. Interestingly, α-SMA+ CAFs were shown to phagocytose tumour-derived antigens and to be enriched in autophagosomes, suggesting an autophagy-dependent handling of those antigens. Indeed, conditional knockout of the autophagy pathway in

α-SMA+ CAFs attenuated tumour development, reduced the infiltration of Treg cells and potentiated the efficacy of immune checkpoint inhibitor (ICI) immunotherapy. Our findings shed light on unknown mechanisms through which CAFs impede anti-tumour immunity and may pave the way for the design of CAF-directed therapies.

## Results

### Partial α-SMA+ CAF depletion promotes tumour regression and contraction of the Treg cell compartment

Recent findings point to increased heterogeneity in the CAF compartment, with distinct molecular signatures suggesting that diverse CAF subsets contribute differently to tumour development[13,16–18,21]. CAFs expressing α-SMA appear as a dominant subpopulation in TME, however their functional importance in regard to anti-tumour immunity, remains controversial. Herein, using αSMA[RFP] reporter mice, we demonstrate that in the course of B16.F10 melanoma cell growth, α-SMA+ CAFs accumulated in the TME and were significantly enriched on Day15 compared to Day10 of tumour growth (Fig. 1a). To shed light on the functional importance of α-SMA+ CAF accumulation during tumour development and induction of anti-tumour immunity, we employed αSMA-tk transgenic mice, which allow for selective in vivo deletion of α-SMA+ CAFs, upon systemic intraperitoneal (i.p.) ganciclovir (GCV) administration[15,24]. GCV treatment did not have an effect on B16.F10 tumour development, since no differences were observed in GCV-treated and control B16.F10-injected wild-type (WT) mice (Supplementary Fig. 1a). Interestingly, i.p. administration of GCV (12.5 mg GCV/kg body weight) every 48 h (Fig. 1b), which resulted in a partial depletion of proliferating (Ki-67+) CD45-EpCAM-CD31- α-SMA+ CAFs (Supplementary Fig. 1b, c) in B16.F10-inoculated αSMA-tk mice, significantly attenuated tumour growth as compared to control-treated animals (Fig. 1c). Tumour regression, following α-SMA+ CAF ablation, was accompanied by decreased frequencies of CD4+ T cells (Fig. 1d, e) while the frequencies of CD11c+ dendritic cells (DCs) and CD11c-CD11b+GR-1+ myeloid-derived suppressor cells (MDSCs) in the TME bore no differences between the two groups (Supplementary Fig. 1d, e). However, both DCs and MDSCs expressed higher levels of the major histocompatibility complex class II (MHC-II) molecule I-A in GCV-treated αSMA-tk mice (Supplementary Fig. 1f), which is associated with a more immunogenic phenotype and the development of a robust anti-tumour immunity[25]. In support, despite the similar frequencies (Fig. 1d, e), tumour infiltrating CD8+ T cells in GCV-treated mice exhibited increased IFN-γ expression compared to control animals, suggesting that α-SMA+ CAF partial depletion led to more efficient cytotoxic responses (Fig. 1f). Interestingly, the frequencies of CD4+Foxp3+ Treg cells, among tumour-infiltrating CD45+ leucocytes, were significantly reduced in GCV-treated compared to PBS-treated B16.F10-inoculated αSMA-tk mice (Fig. 1e). Elaborating more on this finding, we generated αSMA-tk;RFP mice by crossing αSMA-tk and αSMA[RFP] mice. To this end, immunofluorescence in B16.F10 tumour cryosections derived from αSMA-tk;RFP mice confirmed a significant reduction of Foxp3+ Treg cells infiltrating the TME in fibroblast-depleted tumours (Fig. 1g). Overall, these findings demonstrate that partial depletion of α-SMA+ CAFs during B16.F10 melanoma development results in tumour regression and enhanced anti-tumour immunity, characterized by reduced intra-tumoral accumulation of CD4+Foxp3+ Treg cells.

### Foxp3+ Treg cells preferentially accumulate in α-SMA+ CAF-rich areas in the TME

Driven by the observed reduction of Foxp3+ Treg cells in α-SMA+ CAF-depleted tumours, combined with recent findings suggesting that subsets of CAFs are in close proximity with Treg cells[21,26,27], we formulated the hypothesis that α-SMA+ CAFs cross-talk with Foxp3+ Treg cells in the TME, empowering the function of the latter. To investigate this, we first assessed the spatial distribution of the two cell types in

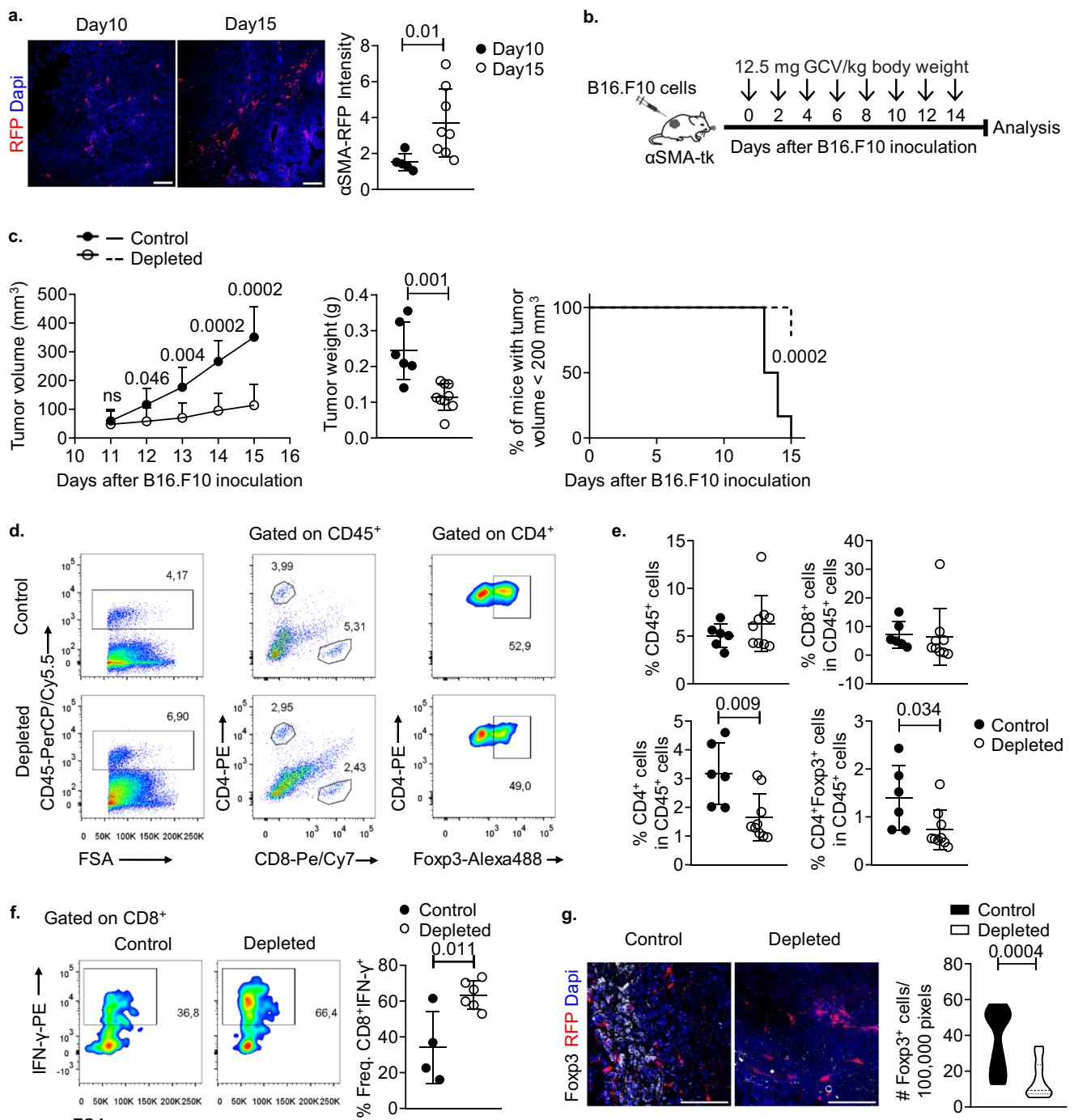

**Fig. 1 | Tumour regression upon partial depletion of α-SMA⁺ CAFs.**
**a** Representative images of αSMA^RFP immunofluorescence and quantification plot of α-SMA intensity from Day10 (n = 2) and Day15 (n = 2) B16.F10 tumour cryosections. At least five fields per tumour were quantified. Scale bar: 100 μm. **b** Outline of Ganciclovir (GCV) treatment in tumour-bearing αSMA-tk⁺ mice. **c** Tumour volume (mm³), tumour weight (g) and percentage of mice bearing tumours <200 mm³ of B16.F10 inoculated PBS-treated (control, n = 6) and GCV-treated (depleted, n = 8) αSMA-tk⁺ mice. **d** and **e** Gating strategy (**d**), percentages (**e**) of intra-tumoral CD45⁺ cells, CD4⁺, CD8⁺ and CD4⁺Foxp3⁺ T cells on Day15 after B16.F10 inoculation of PBS-treated (control, n = 6) and GCV-treated (depleted, n = 9) αSMA-tk⁺ mice. **f** Representative FACS plots and percentages of IFN-γ⁺ cells of intra-tumoral CD8⁺ cells isolated on Day15 after B16.F10 inoculation of PBS-treated (control, n = 4) and

GCV-treated (depleted, n = 6) αSMA-tk⁺ mice. **g** Representative images of αSMA-RFP, Foxp3 immunofluorescence from Day15 B16.F10 tumour cryosections derived from αSMA-tk;RFP mice. At least 5 fields per tumour were captured. Scale bar: 200 μm. Quantitation plots depicting number of Foxp3⁺ cells infiltrating Day15 B16.F10 tumours of PBS-treated (control, n = 3) and GCV-treated (depleted, n = 3) αSMA-tk;RFP mice. Data are shown as mean ± SD. Representative data from four (**c**–**e**), three (**f**) and two (**g**) independent experiments are shown; 6–9 mice/group were used in each experiment for (**c**–**e**) and 2–3 mice/group were used in each experiment for (**g**). Unpaired two-tailed t-test (**c, e**–**g**), Mann-Whitney two-tailed U-test (**a**), two-tailed Log-rank test (**c**). n = biologically independent mouse samples. P values are as indicated in the respective graph. Source data are provided as a Source Data file.

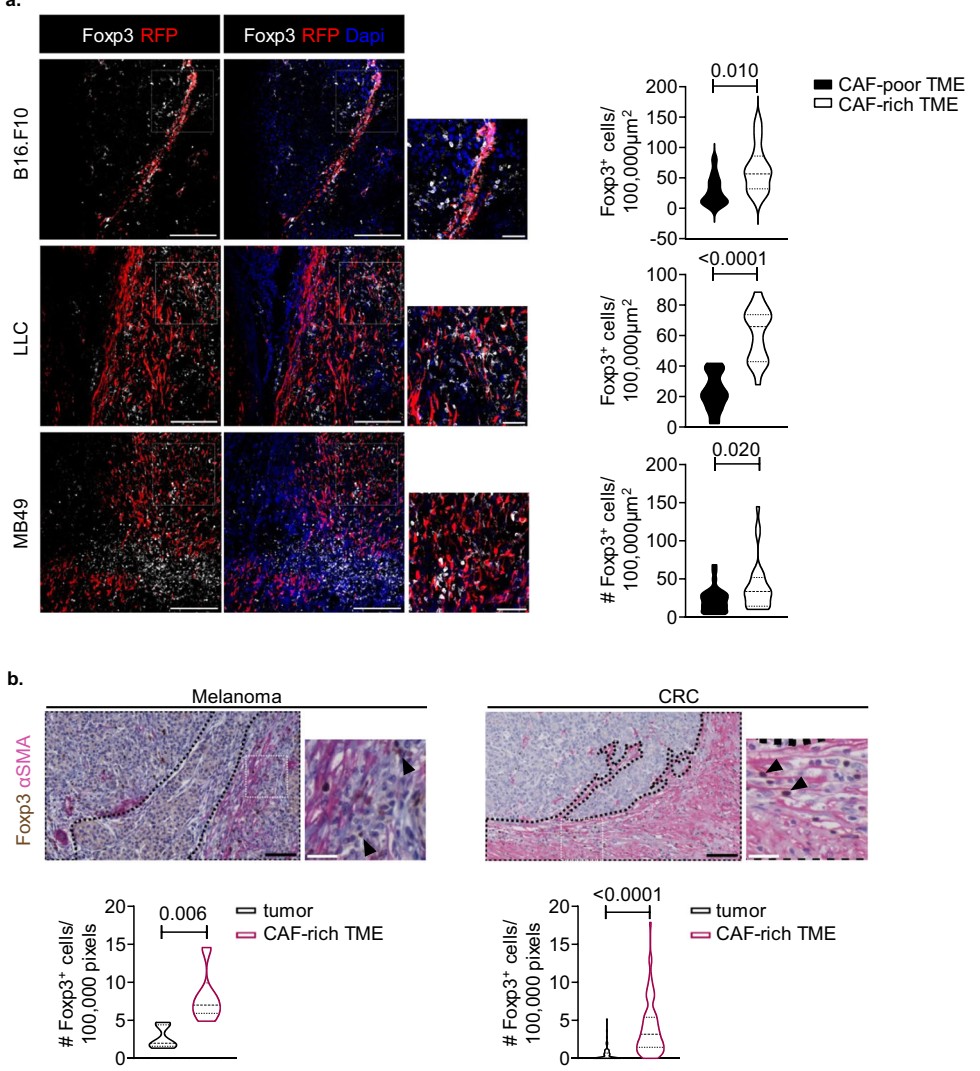

**Fig. 2 | Treg cells accumulate in α-SMA⁺ CAF-rich areas. a** Representative images of αSMA-RFP, Foxp3 immunofluorescence from Day15 B16.F10 (*n* = 3) or Lewis lung carcinoma (LLC) (*n* = 3) or MB49 (*n* = 3) tumour cryosections derived from αSMA-RFP mice. Violin plots depicting the number of Foxp3⁺ cells in CAF-rich and CAF-poor regions. Six fields or more per tumour were captured. Scale bars: 100, 25 μm (inset). **b** Representative images of α-SMA, Foxp3 immunohistochemistry in melanoma (*n* = 3) and colorectal cancer (CRC) (*n* = 15) sections, showing tumour areas and areas enriched in α-SMA⁺ CAFs. Violin plots depicting the number of Foxp3⁺ cells in tumour areas and areas enriched in α-SMA⁺ CAFs. Scale bars: 60.7, 20.3 μm (inset). Paired two-tailed *t*-test (**a**, **b**). *n* = biologically independent mouse samples (**a**); *n* = biologically independent donor samples (**b**). *P* values are as indicated in the respective graph. Source data are provided as a Source Data file.

various tumour models. Immunofluorescence analysis of Day15 tumours from B16.F10-inoculated αSMA^RFP mice revealed that Treg cells preferentially accumulated in α-SMA⁺ CAF-rich areas (Fig. 2a). To examine how this finding relates to the tumour immunogenicity, besides the non-immunogenic B16.F10 melanomas, we also assessed the presence of Foxp3⁺ Treg cells in the TME of αSMA^RFP mice inoculated with the poorly immunogenic Lewis lung carcinoma (LLC) cells or the highly immunogenic bladder carcinoma MB49 cells[28]. Consistent with the results obtained from B16.F10 tumours, in both tumour models, immunofluorescence analyses showed that Foxp3⁺ Treg cells were enriched in areas with abundant α-SMA⁺ CAFs (Fig. 2a). Considering that immune cells preferentially localize in peri-tumoral stromal areas, we next asked whether other T cell subsets (i.e., CD4⁺Foxp3⁻ T effector cells and CD8⁺ cytotoxic T cells) are also in close proximity to α-SMA⁺ CAFs. To this end, T effector cells were also significantly enriched in α-SMA⁺ CAF-rich areas compared to α-SMA⁺ CAF-poor areas of MB49- and LLC-inoculated animals (Supplementary Fig. 2b), while no differences were observed in B16.F10 melanoma animals (Supplementary Fig. 2a). In addition, spatial accumulation of

CD8⁺ T cells was not different between α-SMA⁺ CAF-rich and α-SMA⁺ CAF-poor areas across the three tumour mouse models (Supplementary Fig. 2c). Of interest, immunohistochemical analyses of tumour specimens derived from 15 stage II and stage III colorectal carcinoma (CRC) patients and 3 melanoma patients demonstrated that Foxp3⁺ Treg cells are located in areas with stromal cells abundantly expressing α-SMA (Fig. 2b) as compared to tumour cell areas. Taken together, these results indicate that Foxp3⁺ Tregs preferentially accumulate in α-SMA⁺ CAF enriched areas as evident in both mouse and human tumours and raise the possibility of a functional cross-talk between the two cell subsets.

## α-SMA⁺ CAFs form an immunological synapse with Foxp3⁺ Treg cells in the TME

To support a functional cross-talk with Foxp3⁺ Treg cells, α-SMA⁺ CAFs should possess antigen uptake, processing and presenting capacities. Thus, we first examined the ability of α-SMA⁺ CAFs to uptake tumour antigens. To this end, α-SMA⁺ CAFs isolated from tumours of B16.F10-inoculated mice were stained with cyto-ID (red) dye and co-cultured

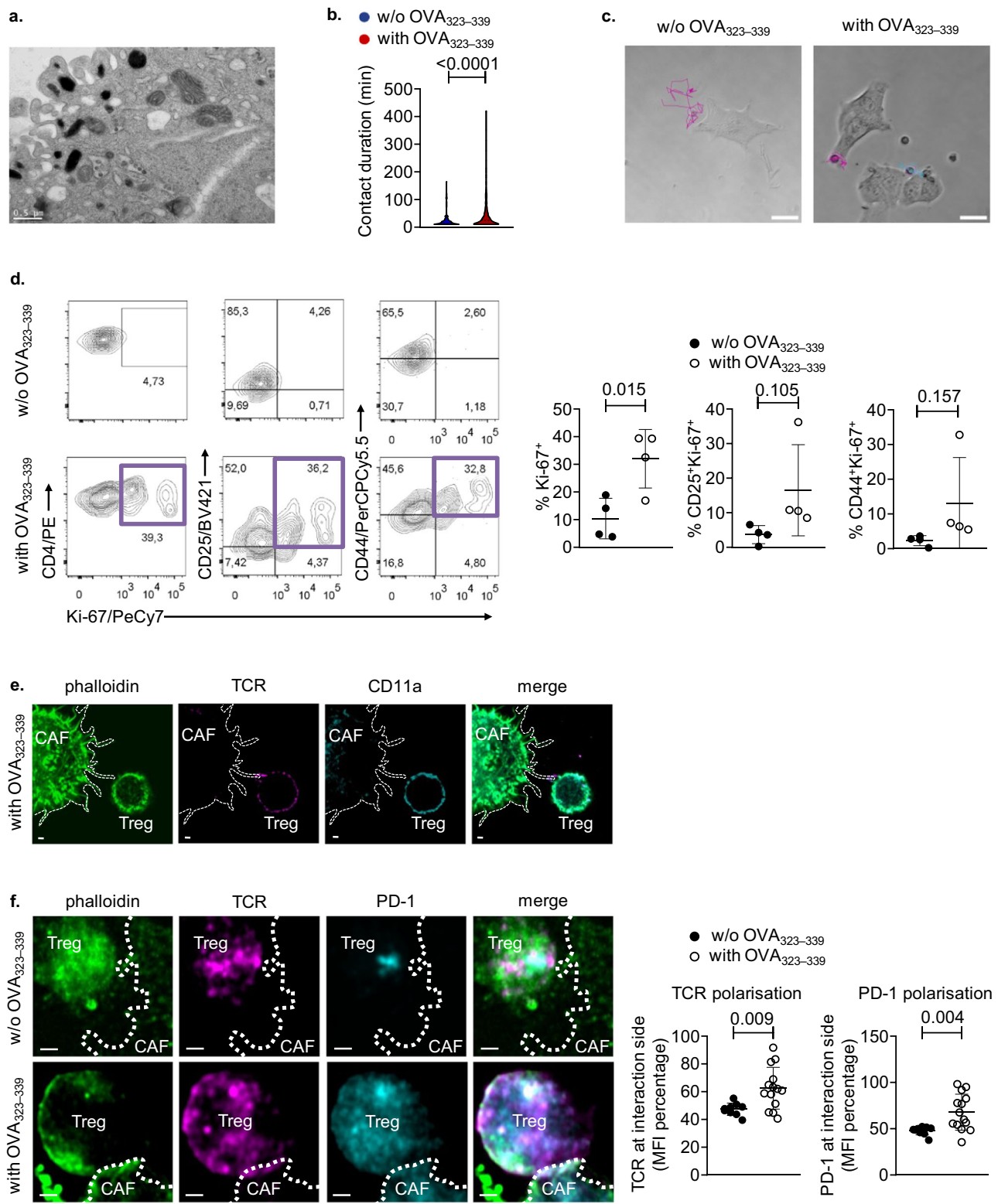

with CMFDA (green) stained B16-F10 melanoma cells either untreated or cisplatin-treated to induce cell death (Supplementary Fig. 3a). Flow cytometry analysis revealed that $\alpha$-SMA$^+$ CAFs phagocytosed both untreated and cisplatin-treated melanoma antigens (Supplementary Fig. 3b). Interestingly, electron microscopy analysis of $\alpha$-SMA$^+$ CAFs from B16.F10 melanoma injected mice, showed extensive membrane protrusions, a characteristic of cells undergoing phagocytosis, as well as the presence of melanosome-like structures close to the cell membrane (Fig. 3a), proposing that $\alpha$-SMA$^+$ CAFs could uptake tumour antigens in the TME. The antigen processing ability of $\alpha$-SMA$^+$ CAFs was directly assessed upon co-culture with the DQ-OVA, a self-quenched conjugate of ovalbumin that exhibits fluorescence upon proteolytic degradation into dye-labelled peptides. As shown in Supplementary Fig. 3c, CAFs exhibited bright fluorescence upon culture with DQ-OVA which was reduced in the presence of NH$_4$Cl lysosomal inhibitor. Furthermore, ex-vivo isolated $\alpha$-SMA$^+$ CAFs showed to express MHC class I and MHC class II molecules as well as low levels of co-stimulatory molecules CD80 and CD86 and the co-inhibitory

**Fig. 3 | α-SMA⁺ CAFs form synapses with Treg cells in an antigen-specific manner. a** Transmission electron microscopy image of isolated CAFs. Scale bar: 0.5 μm. **b** Violin plots representing contact duration between OTII Treg cells and CAFs in the absence or presence of OVA$_{323-339}$. **c** Representative trajectories of individual OTII Treg cells under co-culture conditions in the absence (left) or presence (right) of OVA$_{323-339}$. Scale bar: 0.65 μm. **d** Flow cytometry plots (left), percentages (right) of Ki-67⁺, CD25⁺Ki-67⁺, and CD44⁺Ki-67⁺ OTII Treg cells following co-culture with isolated CAFs and IL-2 in the absence ($n = 4$) or presence ($n = 4$) of OVA$_{323-339}$. **e** and **f** Representative maximum projections of 3D fluorescent confocal microscopic image stacks of CD45⁻CD31⁻EpCAM⁻α-SMA⁺ CAFs and OTII Tregs stained for actin, TCR and LFA-1α, in the presence of OVA$_{323-339}$ or stained for actin, TCR, and PD-1, in presence and absence of OVA$_{323-339}$ after 15 min of co-culture. Scale bar: 1 μm. Quantification plots of TCR and PD-1 (mean fluorescence intensity, MFI) near the interface between Treg and CAF-extending actin-rich filopodia; interface MFI was normalized to mean intensity of each respective channel in the cell of interest (MFI percentage). Data are shown as mean ± SD. Representative data from four (**b**–**d**) and one (**e**, **f**) independent experiments are shown; CAFs isolated from Day 15 tumours of 1 mouse were used in each experiment for (**b**, **c**) and CAFs from 2 to 4 mice were used in each experiment for (**d**–**f**). Unpaired two-tailed *t*-test (**b**, **d**), Mann–Whitney two-tailed *U*-test (**f**), $n$ = biologically independent mouse samples. *P* values are as indicated in the respective graph. Source data are provided as a Source Data file.

molecules PD-L1 and PD-L2 (Supplementary Fig. 3d), consistent with a tolerogenic phenotype[25,29]. By contrast, CD45⁺CD11b⁺ and CD45⁺CD11c⁺ myeloid cells showed enhanced expression of the aforementioned molecules, while CD45⁻CD31⁺ endothelial cells expressed low or not-detectable levels (Supplementary Fig. 3d).

To examine the functional importance of these findings, we performed in vitro co-culture of CD4⁺CD25⁺GITR⁺Vα2⁺Vβ5.1⁺ Treg cells, isolated from OTII TCR transgenic mice, with OVA$_{323-339}$ pulsed α-SMA⁺ CAFs (Supplementary Fig. 4a). First, we sought to determine the dynamic interactions between CAFs and Treg cells, and for this reason we performed time-lapse video microscopy during 8–12 h of co-culture between the two cell types (Supplementary movies 1 and 2). We developed an automated methodology to determine Treg cell trajectories over time and to quantify the length of interactions between the two cell types. As shown in Supplementary Fig. 4b, a movement arrest was observed between OVA$_{323-339}$ loaded CAFs and Treg cells, indicating synapse formation. Analysis of the acquired videos revealed that Treg cells formed longer interactions with CAFs which were pulsed with OVA$_{323-339}$ antigen (Fig. 3b, Supplementary movies 1 and 2). Also, Treg cell trajectories were significantly smaller in the presence of OVA peptide, as Treg cells were retained in contact with CAFs for considerably longer periods of time (Fig. 3c). Furthermore, following the same experimental setup, we addressed the effect of antigen presentation by CAFs in Treg cell functional features. As shown in Fig. 3d, OTII Treg cells exhibited increased proliferation, based on Ki-67 expression, in the presence of OVA$_{323-339}$ pulsed α-SMA⁺ CAFs compared to non-antigen pulsed CAFs. As a positive control, OTII Tregs were stimulated with anti-CD3/anti-CD28 beads. The enhanced proliferation of Treg cells, observed in the presence of OVA$_{323-339}$ stimulation, was accompanied by enhanced activation, as shown by increased CD25 and CD44 expression by peptide-stimulated OTII Treg cells, compared to Treg cells co-cultured with non-pulsed CAFs. (Fig. 3d). Remarkably, co-culture of OVA$_{323-339}$ stimulated α-SMA⁺ CAFs with OTII T effector cells (Teffs, identified as CD4⁺CD25⁻GITR⁻Vα2⁺Vβ5.1⁺ cells) illustrated that α-SMA⁺ CAFs cannot support the activation, nor proliferation of the latter cell type (Supplementary Fig. 4c). These results confirm that α-SMA⁺ CAFs have the capacity to uptake, process and present tumour antigens to promote the activation and proliferation of Foxp3⁺ Treg cells, raising the possibility of immunological synapse formation between the two cell subsets.

To provide direct evidence of this hypothesis, we first confirmed the expression of molecules, which constitute integral components of the immunological synapse, by α-SMA⁺ CAFs such as ICAM-1 and CD48 (Supplementary Fig. 5a)[30]. Notably, the co-culture of OTII Treg cells and OVA$_{323-339}$-pulsed α-SMA⁺ CAFs (Supplementary Fig 4a) demonstrated that the interaction between the two cell types followed the criteria that define an immunological synapse[31]. Specifically, we found that only in the presence of OVA$_{323-339}$, Foxp3⁺ Treg cells and α-SMA⁺ CAFs communicate on filopodia extending from the latter (Supplementary movie 3), suggesting adhesive interactions. Wrapping of Treg cells around the actin protrusions formed by α-SMA⁺ CAFs (Supplementary movie 3) pointed to the stability of the synapse. Furthermore,

staining for TCR/CD11a depicted that Treg cells and OVA$_{323-339}$ pulsed α-SMA⁺ CAFs were not fused, yet retained their individuality (Fig. 3e). Finally, TCR/PD-1 staining revealed that there is enhanced polarization in the synapse location, hinting to polarized secretion (Fig. 3f). In support of these data, confocal microscopy analysis of tumour tissues revealed a co-localization between α-SMA⁺ CAFs and Foxp3⁺ Tregs (Supplementary Fig. 5b), indicative of synapse formation between the two cell types in the TME. Overall, the findings presented here illustrate that α-SMA⁺ CAFs form an immunological synapse with Foxp3⁺ Treg cells in the TME and instruct their activation and proliferation in an antigen-specific manner.

## Autophagy and antigen presentation pathways are enriched in α-SMA⁺ CAF clusters

Breaking down the α-SMA⁺ CAF/Foxp3⁺ Treg cell synapse may represent a potential immunotherapeutic target in cancer and thus, we sought to shed light on mechanisms operating in the synapse formation. We performed a meta-analysis of published single-cell transcriptomic datasets, which investigated mesothelial cell-associated cell populations from 4T1 murine breast carcinomas[18], PDAC tumours from *Kras*$^{+/LSL-G12D}$; *Trp53*$^{+/LSL-R172H}$; *Pdx1*$^{cre}$ (KPC) mice[16] and B16.F10 melanomas[32]. Annotation of CAFs was based on the expression of cell marker genes as referenced in Grauel et al.; specifically, CAFs were defined as cells lacking expression of epithelial marker gene *Epcam*, endothelial marker *Pecam1*, hematopoietic cell marker *Ptprc* and expressing mesenchymal and fibroblast marker genes such *as Col1a1, Col3a1, Thy1, Pdpn*, and *Fap*[18]. To identify the fibroblast clusters in these datasets, we performed graph-based clustering and generated uniform manifold approximation and projection (UMAP) plots. In Grauel et al.[18] and Davidson et al.[32] datasets (Fig. 4a), 8 distinct cell clusters were identified, while in Elyada et al.[16] dataset, we identified 6 clusters (Fig. 4a) and found that in all datasets, clusters 3 and 5 expressed the fibroblast-specific genes in addition to *Acta2* (encoding for α-SMA) (Supplementary Figs. 6a, 7a, and 8a); hence we identified these two subsets as α-SMA⁺ expressing CAFs. Importantly, re-analysis of the datasets showed that clusters 3 and 5 exhibited enrichment in genes associated with MHC class II antigen presentation, such as *Cd74*, which encodes for the invariant chain[33] and the MHC class II encoding genes *H2-Aa, H2-DMa* and *H2-DMb1*[34] (Fig. 4b), in agreement with studies addressing the role of specific CAF subsets acting as non-professional antigen-presenting cells. Strikingly, α-SMA⁺ CAFs were revealed to upregulate the expression of genes associated with the autophagic machinery (Fig. 4c), a pathway which is induced under the hypoxic conditions of the TME and contributes in the pool of peptides displayed via MHC class II molecules[25,35,36]. Most notable examples included *Wipi1*, which plays a key role in autophagosome biogenesis, the lysosomal-associated membrane protein 2 (*Lamp2*) gene, as well as, *Gabarapl1* and *Sting1*, which act as positive regulators of the autophagic process in tumour settings[37,38] (Fig. 4c). Collectively, these data are in accordance with our findings that α-SMA⁺ CAFs may act as non-professional antigen-presenting cells, which are found to be characterized by increased autophagy.

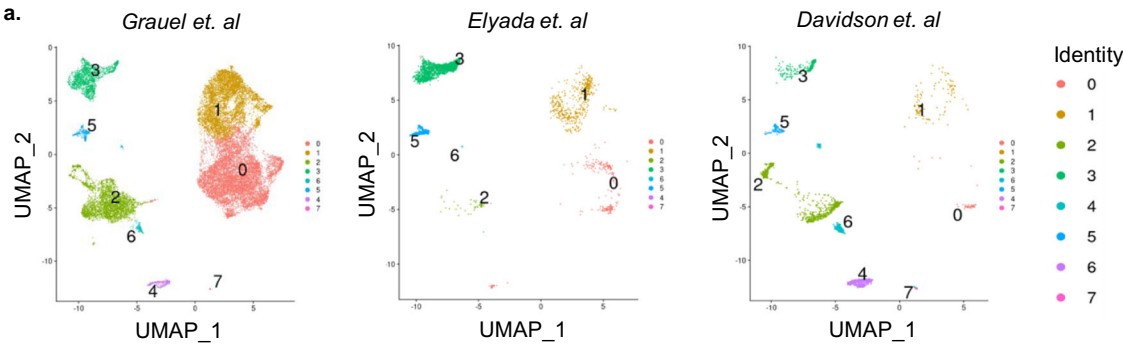

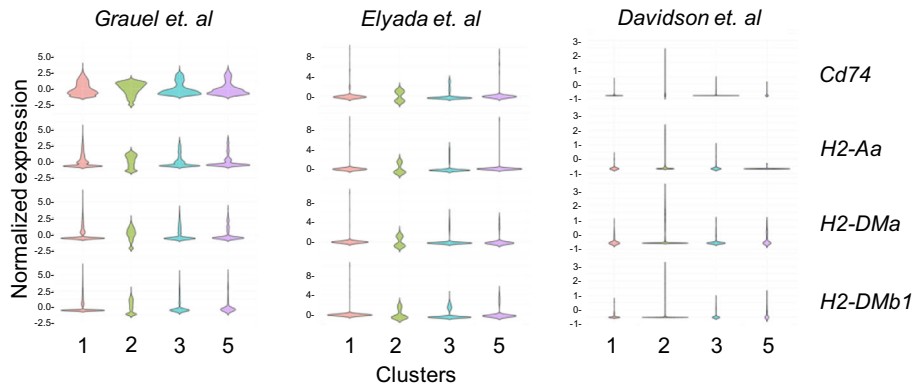

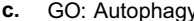

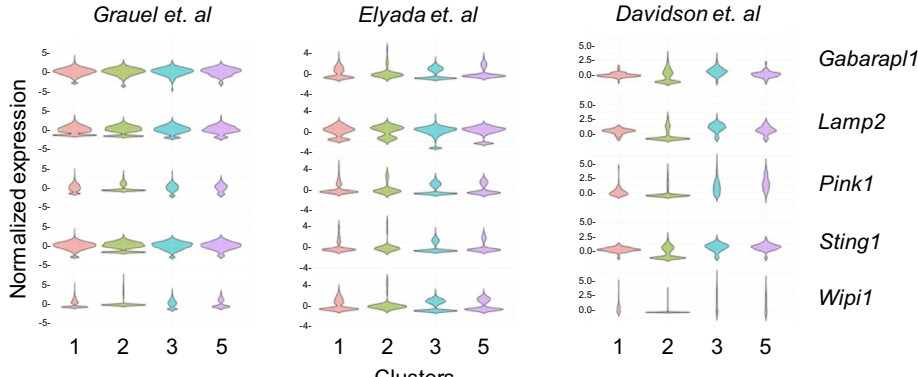

**Fig. 4 | Enrichment of autophagy and antigen processing/presentation pathways in α-SMA⁺ CAFs. a** Graph-based clustering of cells with uniform manifold approximation and projection (UMAP) was performed; 8 clusters were identified. **b** and **c** Violin plots demonstrating genes associated with GO: Antigen processing and presentation of peptide or polysaccharide antigen via MHC class II (**b**) and GO: Autophagy (**c**) for clusters 1, 2, 3, 5 identified from Grauel et al., Elyada et al. and Davidson et al. data. The width of the violin plot represents the frequency of cells. Values on the *y*-axis indicate normalized gene expression.

## Ablation of autophagy in CAFs diminishes Treg cell accumulation and promotes tumour regression

Consistent with the scRNA-seq data, analysis of α-SMA⁺ CAF electron microscopy revealed a prominent presence of double-membrane structures containing cargo or being in the fusion process with lysosomes (Fig. 5a and Supplementary Fig. 9a), resembling autophagosomes (Fig. 5a, arrows). To provide evidence for functional autophagy in α-SMA⁺ CAFs, we performed a flux assay to measure the levels of cellular autophagic degradation activity in vitro by confocal microscopy. Thus, upon treatment with NH₄Cl, a lysosomal degradation inhibitor that blocks the fusion of autophagosomes with the lysosomes, α-SMA⁺ CAFs isolated from B16.F10 melanomas exhibited increased accumulation of the autophagosomal microtubule-associated protein 1A-1B-light chain 3 (LC3), which is essential for the formation of autophagosomes[25], as well as of the adaptor protein Sequestosome-1 (SQSTM1/p62), which directly binds to LC3 and targets the ubiquitinated proteins that are supposed to be degraded via the autophagy machinery[25]. The accumulated proteins were also highly co-localized with the lysosomal-associated membrane protein 1 (Lamp-1) in NH₄Cl-treated CAFs compared to vehicle CAFs, denoting the formation of functional autophagolysosomes in CAFs (Fig. 5b).

Since autophagy has an established role in antigen processing and presentation, we sought to examine whether ablation of autophagy in α-SMA⁺ CAFs inhibits the formation of synapse with Foxp3⁺ Treg cells,

promoting tumour regression. To this end, we generated the conditional knock-out mice α-SMA^creAtg5^fl/fl by crossing α-SMA^cre with Atg5^fl/fl mice (Supplementary Fig. 9b). Real-time PCR analysis confirmed the

deletion of Atg5 gene in α-SMA⁺ CAFs (Supplementary Fig. 9c), which associated with decreased levels of autophagosome formation based on LC3 expression (Supplementary Fig. 9d). In steady state,

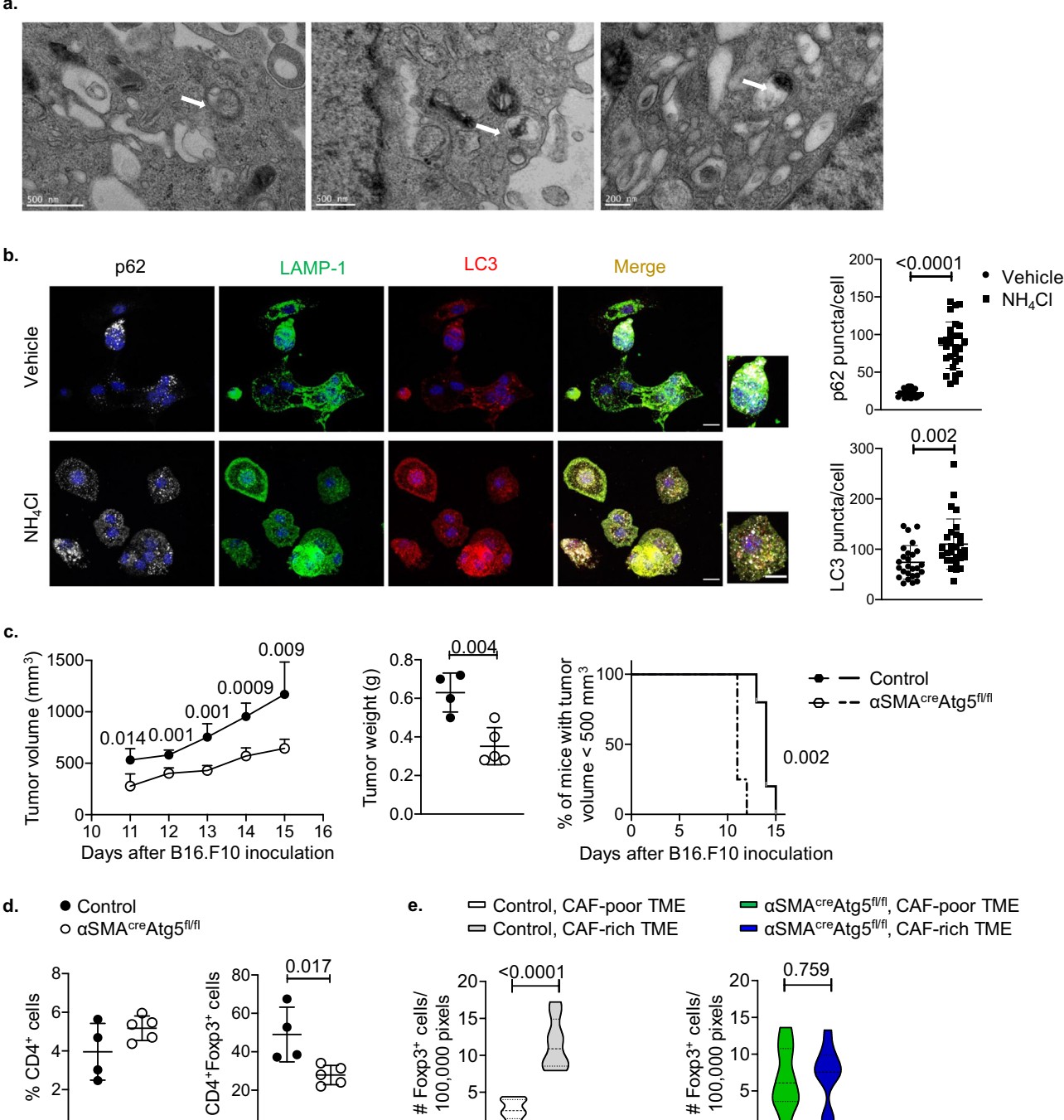

**Fig. 5 | Conditional ablation of autophagy in α-SMA⁺ CAFs promotes tumour regression. a** Transmission electron microscopy images of CAFs isolated from Day15 B16.F10 melanomas. Scale bar: 500 or 200 nm, as indicated in the figure. **b** Autophagic flux assay. Representative images of immunofluorescence confocal microscopy for LC3 (red), LAMP-1 (green), p62 (silver white), and DAPI (blue) in NH₄Cl-treated CAFs (*n* = 28) and vehicle-treated CAFs (*n* = 26). CAFs were isolated from Day15 B16.F10 melanomas. Scale bar: 12; 20 μm (inset). p62 puncta/cell and LC3 puncta/cell and are depicted. **c** Tumour volume (mm³), tumour weight (g) and percentage of mice bearing tumours <500 mm³ of B16.F10 inoculated Atg5^fl/fl (*n* = 4) and α-SMA^creAtg5^fl/fl (*n* = 5) mice. **d** Percentages of intra-tumoral CD4⁺ T cells and CD4⁺Foxp3⁺ T cells on Day15 after B16.F10 inoculation of Atg5^fl/fl (*n* = 4) and α-SMA^creAtg5^fl/fl (*n* = 5) mice. **e** αSMA, Foxp3 immunofluorescence from Day15

B16.F10 tumour cryosections derived from control (*n* = 2) and α-SMA^creAtg5^fl/fl (*n* = 2) mice. At least 5 fields per tumour were captured. Quantitation plots depicting a number of Foxp3⁺ cells infiltrating Day15 B16.F10 tumours in CAF-poor and CAF-rich regions of control (*n* = 2) and α-SMA^creAtg5^fl/fl (*n* = 2) mice. Data are shown as mean ± SD. Representative data from one (**a**), two (**e**), three (**b**), five (**c**, **d**) independent experiments are shown; CAFs isolated from Day15 tumours of 3 mice were used in each experiment for (**b**), 4–8 mice/group were used in each experiment for (**c**, **d**) and 2 mice/group were used in each experiment for (**e**). Unpaired two-tailed *t*-test (**b**–**d**), paired two-tailed *t*-test (**e**). ns = non-significant. *n* = number of CAFs counted (**b**); *n* = biologically independent mouse samples (**c**–**e**). *P* values are as indicated in the respective graph. Source data are provided as a Source Data file.

α-SMA$^{cre}$Atg5$^{fl/fl}$ mice did not exhibit any sign of intestinal or lung inflammation (Supplementary Fig. 9e) and frequencies of CD4$^+$, CD8$^+$ T lymphocytes in the LNs as well as of myeloid cell subsets were similar between the two groups of mice with the exception of Treg cell frequencies which were increased in the LNs of α-SMA$^{cre}$Atg5$^{fl/fl}$ mice (Supplementary Fig. 9f). Interestingly, α-SMA$^{cre}$Atg5$^{fl/fl}$ exhibited a significant reduction of tumour growth upon inoculation with B16.F10 melanoma cells, compared to Atg5$^{fl/fl}$ control mice (Fig. 5c). Furthermore, tumour regression in α-SMA$^{cre}$Atg5$^{fl/fl}$ mice was accompanied by significantly decreased frequencies of tumour infiltrating Foxp3$^+$ Treg cells (Supplementary Fig. 10, Fig. 5d). In addition, Treg cells lost their preference to accumulate in α-SMA$^+$ CAF-rich areas in the TME of α-SMA$^{cre}$Atg5$^{fl/fl}$ mice (Fig. 5e), pinpointing to a disturbance in the synapse formation between α-SMA$^+$ CAFs and Treg cells upon autophagy depletion. Finally, autophagy ablated α-SMA$^+$ CAFs, pulsed with the entire OVA antigen, did not support the activation and proliferation of OT II Foxp3$^+$ Treg cells as compared to α-SMA$^+$ CAFs from control mice (Supplementary Fig. 11), suggesting that they lost their ability to process and present antigens. Collectively, these data propose that ablation of autophagy in α-SMA$^+$ CAFs hampers tumour progression and hinders the cross-talk between α-SMA$^+$ CAFs and Treg cells in an antigen-dependent fashion, also curtailing Treg infiltration in the tumour stroma.

## Ablation of autophagy reprograms CAFs into a pro-inflammatory phenotype

To gain molecular insights on mechanisms used by α-SMA$^+$ CAF autophagy to dictate the anti-tumour immunity and alter their capability to process tumour antigens to Foxp3$^+$ Treg cells, we performed a transcriptomic and proteomic analysis in CAFs isolated from α-SMA$^{cre}$Atg5$^{fl/fl}$ vs. control tumour-inoculated mice, including analysis of CAF secretome. To this end, autophagy-ablated CAFs demonstrated extensive transcriptional alterations, with 1332 differentially expressed genes compared to CAFs isolated from Atg5$^{fl/fl}$ littermate control animals (Fig. 6a, log$_2$(fold change) > 1.5 and adjusted $P$-value < 0.05). Gene expression analysis showed a reshaped transcriptional profile in autophagy-deficient CAFs. Specifically, CAFs isolated from α-SMA$^{cre}$Atg5$^{fl/fl}$ littermates were characterized by downregulation of genes associated with the respiratory chain and oxidative phosphorylation (e.g. *Cox8a, Cox7c, Ndufa1, Cox7a2, Atp5l, Cox16, Atp5j, Atp5md, Uqcr11*, Fig. 6b), indicating that by the absence of autophagy CAFs may rely further on aerobic glycolysis to support their metabolic needs, enhancing the "Warburg effect" which is already a cardinal feature of CAFs in the TME[5]. Also, autophagy-ablated CAFs downregulated several genes involved in the PI3K/Akt/mTOR signalling pathway (e.g., *Foxo4, Lamtor2, Akt1, Akt1s1, Pik3r3, Lamtor1*, Fig. 6b), which acts as a central regulator of cell metabolism, including the autophagic machinery[39], further pinpointing to an altered metabolic phenotype. In addition, proteomic analysis of α-SMA$^+$ CAF cell lysates revealed a total of 1781 differentially expressed proteins (DEPs) upon depletion of autophagy (Supplementary Fig. 12b, left). Similar to the transcriptomic data, enrichment analysis showed a downregulation of processes associated with oxidative phosphorylation, while among the most robustly enriched terms were glucose metabolism and aerobic respiration (Fig. 6e).

Importantly, CAFs from α-SMA$^{cre}$Atg5$^{fl/fl}$ mice upregulated several cytokines, chemokines and their cognate receptors (e.g., *Cxcl5, Cxcl16, Cxcl13, Il34, Il18, Il18ra, Il6, Ifngr1, Ifnar1*, Fig. 6b), hinting to a shift toward a pro-inflammatory phenotype upon ablation of autophagy. Consistent with the differential expression analysis, gene set enrichment analysis (GSEA) demonstrated that CAFs isolated from α-SMA$^{cre}$Atg5$^{fl/fl}$ mice were found enriched in processes such as cytokine production, C−X−C chemokine receptor activity, C−C chemokine receptor activity, and cytokine production involved in inflammatory response (Fig. 6c, d). Additionally, an enhanced antigen presentation

capacity via the MHC class II complex was also identified (e.g., terms "Positive regulation of MHCII biosynthetic process", "Endocytosis", "Phagocytosis", Fig. 6c, d), suggesting a possible switch from tolerogenic to inflammatory phenotype. The inflammatory nature of α-SMA$^+$ CAFs upon genetic ablation of *Atg5* was further confirmed upon proteomic analysis. Thus, several proteins associated with inducing cellular inflammatory features were found in greater abundance in α-SMA$^+$ CAFs with depleted autophagy (Supplementary Fig. 12b and c, right), while the secretome showed a greater abundance of proteins associated with the complement cascade (e.g., C4b, C1qbp, C2, Pros1) as well as TGF-β signaling pathway (e.g., Cd109, Thbs1) (Supplementary Fig. 12c, right). Also, terms such as "Neutrophil degranulation", "Cytokine signaling in the immune system", "IL-1 signaling" and "MHC class II antigen presentation" (Fig. 6e) were among the most enriched when comparing autophagy-ablated CAFs and CAFs isolated from Atg5$^{fl/fl}$ littermate control animals, showcasing the inflammatory potential of these cells upon autophagy depletion. Overall, our -omics analyses reveal a robust reprogramming in CAFs lacking autophagy, which strongly hints that they acquire more immune-permissive characteristics.

## Autophagy ablation in CAFs potentiates the efficacy of immune checkpoint blockade therapy

Immunotherapy with immune checkpoint inhibitors induces durable responses in a small proportion of patients due to resistance mechanisms that operate in the TME. Since α-SMA$^+$ CAFs express high levels of PD-L1/PD-L2 (Supplementary Fig. 3d and Supplementary Fig. 5a), we asked whether anti-PD-L1 immunotherapy could target the synapse formation with Treg cells. To this end, treatment of B16.F10-inoculated α-SMA-RFP animals with anti-PD-L1 every three days following tumour induction slightly reduced the growth of melanomas (Supplementary Fig. 13a), but without reaching statistical significance, suggesting the involvement of resistance mechanisms as previously proposed[40]. Notably, the numbers of tumour infiltrating Foxp3$^+$ Treg cells that were in proximity to α-SMA$^+$ CAFs were similar between anti-PD-L1- and control-treated animals (Supplementary Fig. 13b), indicating that anti-PD-L1 treatment does not interrupt the synapse formation between CAFs and Treg cells. Interestingly, genetic disturbance of the PD-L1/PD-1 axis, via the use of PD-1$^{-/-}$ animals, showed reduced accumulation of Treg cells in α-SMA$^+$ CAF-rich areas in the TME (Supplementary Fig. 13c), confirming the involvement of the PD-L1/PD-1 axis in the synapse formation between the two cell populations.

We then asked whether α-SMA$^+$ CAF autophagy may exert a synergistic effect with anti-PD-1 and anti-CTLA-4 ICI immunotherapy, which constitute an established therapeutic protocol. To address this, B16.F10-injected α-SMA$^{cre}$Atg5$^{fl/fl}$ or Atg5$^{fl/fl}$ littermate mice were administered with a combination of the immune checkpoint inhibitors anti-PD-1 and anti-CTLA-4, every three days following tumour inoculation (Fig. 7a). Importantly, α-SMA$^{cre}$Atg5$^{fl/fl}$ tumour-bearing mice that received combinational ICI developed significantly reduced tumours compared to ICI-treated control mice or PBS-treated α-SMA$^{cre}$Atg5$^{fl/fl}$ littermates (Fig. 7b). When examining the immune infiltrate, we found remarkable changes in the TME of α-SMA$^{cre}$Atg5$^{fl/fl}$ tumour-bearing mice upon dual ICI. Specifically, the regression of tumour growth in ICI-treated α-SMA$^{cre}$Atg5$^{fl/fl}$ was accompanied by a marked increase of the tumour-infiltrating CD45$^+$ leucocytes, while considering the T cell compartment, a significant expansion of total CD4$^+$ and CD8$^+$ T cells and effector (CD4$^+$Foxp3$^-$) T cells was observed, with notably reduced infiltration of (CD4$^+$Foxp3$^+$) Tregs. In the myeloid compartment, the frequency of tumour-infiltrating DCs (CD11c$^+$) was elevated compared to the ICI-treated control group, whereas the frequency of the immunosuppressive population MDSCs was significantly decreased in ICI-treated α-SMA$^{cre}$Atg5$^{fl/fl}$ mice (Fig. 7c). These alterations suggest a shift towards an effector anti-tumour immune landscape capable of diminishing tumour progression. Overall, these data demonstrate that

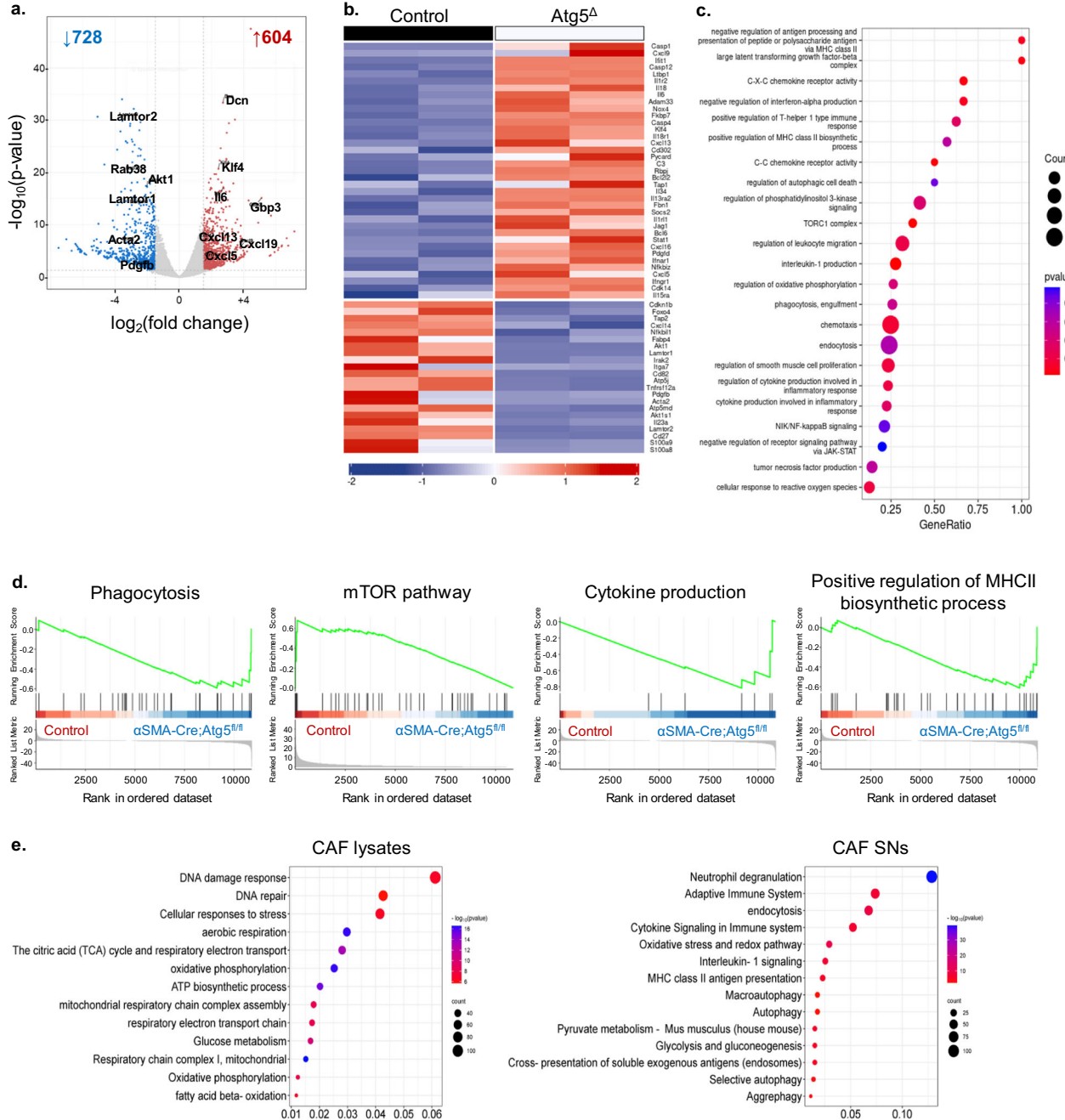

**Fig. 6 | Deficiency of autophagy reprograms α-SMA⁺ CAFs towards an "inflammatory" phenotype. a** Volcano plot showing distribution p-value (−log₁₀ p-value) and fold change (log₂ fold change) distribution of genes identified in CAFs isolated from αSMAcre (n = 2) and αSMAcreAtg5fl/fl (n = 2) mice. **b** Heatmap with scaled expression values (row z-score) of selected differentially expressed genes (DEGs) between αSMAcre (n = 2) and αSMAcreAtg5fl/fl (n = 2) mice. Gene names are depicted on the heatmap. For DEG analysis, the thresholds FDR < 0.05 and −1.5 < log₂FC < 1.5 were used. **c** Bubble plot of enriched pathways determined from transcriptomic data. The

size of the dot represents gene count, and the colour represents the p-value. **d** Gene set enrichment analysis (GSEA) plots showing the enrichment of "Phagocytosis" (NES 1.305275675, FDR 0.517387464), "mTOR signalling" (NES 1.372658556, FDR 0.288520886), "Cytokine production" (NES 1.279629696, FDR 0.000037360), "Positive regulation of MHC II biosynthetic process" (NES −0.816304866, FDR 0.028912909) gene sets. **e** Bubble plots of enriched pathways determined from proteomic data of CAFs lysates (left) and CAF supernatants (right, SNs). The size of the dot represents protein count, and the colour represents the −log₁₀(p-value).

---

the inhibition of autophagy in α-SMA⁺ CAFs acts synergistically with ICI to heighten the anti-tumour immune responses and abrogate the infiltration of Treg cells, leading to tumour regression.

## Discussion

CAFs constitute the most abundant non-hematopoietic cell subset in the TME, which possesses a fundamental role in all tumour facets, including tumour growth, angiogenesis, metastasis as well as response to therapy[5,12,13]. Despite the overwhelming data which establish a

phenotypic and functional heterogeneity of CAFs, based on advancements in single-cell technologies, still the mechanisms via which CAFs shape the anti-tumour immune responses remain poorly understood. Our data reveal a synapse formation between α-SMA⁺ CAFs and Treg cells which endows tumour immune evasion. α-SMA⁺ CAFs exhibit a tolerogenic phenotype, with low expression of MHC II and costimulatory molecules; moreover, they express PD-L1/2 co-inhibitory receptors and promote intra-tumoral Treg cell activation and proliferation in an antigen-specific manner. Of interest, α-SMA⁺ CAFs are

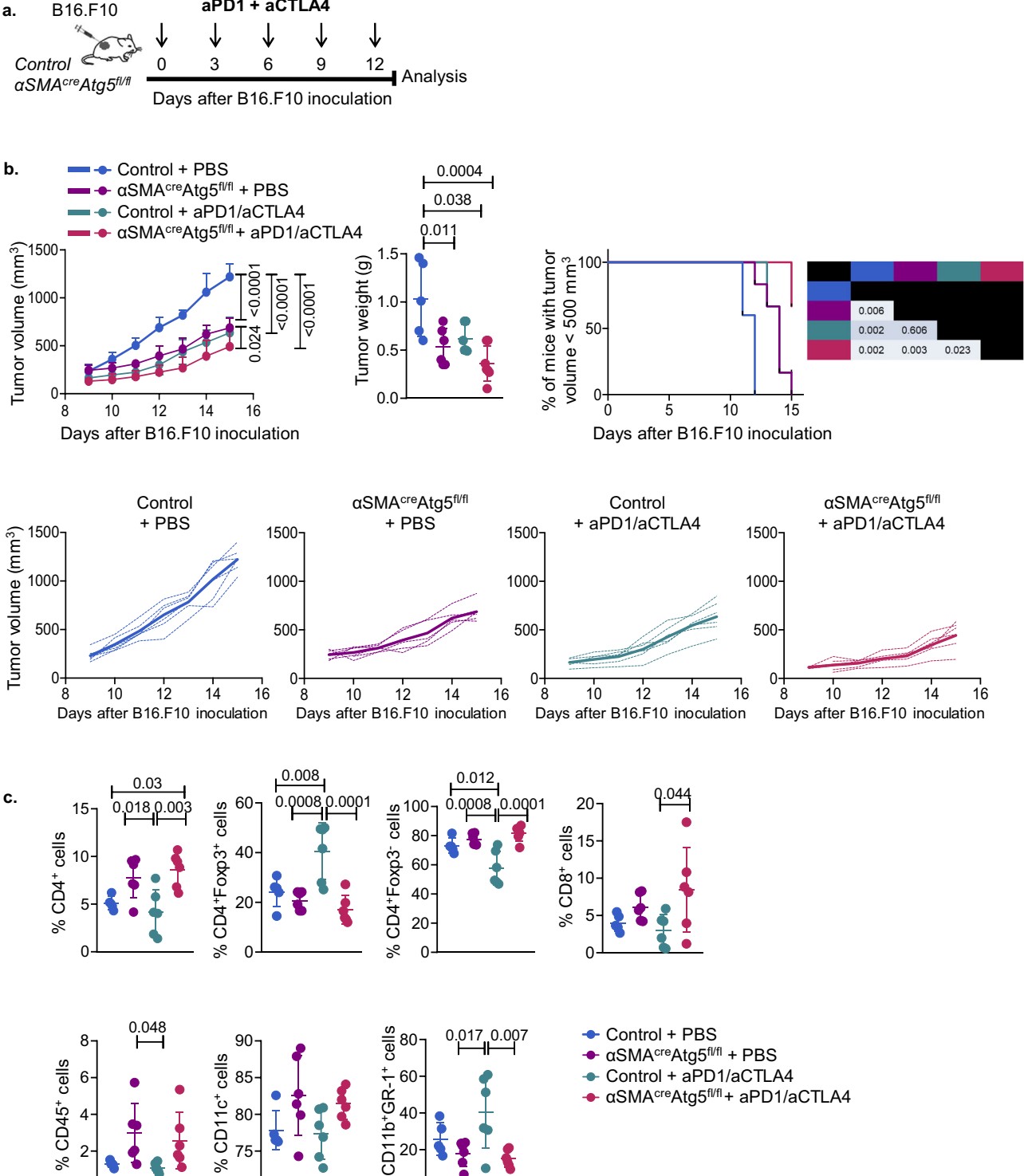

**Fig. 7 | Autophagy deficiency in α-SMA⁺ CAFs potentiates ICI immunotherapy.**
**a** Timeline of tumour progression following combinational administration of anti-PD-1 and anti-CTLA-4 in B16.F10 melanoma bearing C57BL/6 (control) and αSMA$^{cre}$Atg5$^{fl/fl}$ mice. Mice were administered with dual ICI every 3 days after B16.F10 inoculation and were sacrificed on Day15. **b** Tumour volume (mm³), tumour weight (g) and percentage of mice bearing tumours <500 mm³ of B16.F10 inoculated PBS-treated C57BL/6 mice ($n = 5$), PBS-treated αSMA$^{cre}$Atg5$^{fl/fl}$ mice ($n = 6$), anti-PD-1/anti-CTLA-4-treated C57BL/6 mice ($n = 5$), anti-PD-1/anti-CTLA-4-treated αSMA$^{cre}$Atg5$^{fl/fl}$ mice ($n = 6$). Values in the table represent $P$-values between groups in Kaplan−Meier curves. **c** Percentages of intra-tumoral CD45⁺ cells, CD8⁺ T cells,

CD4⁺Foxp3⁻ T cells, CD4⁺Foxp3⁺ T cells, CD11c⁺ DCs and CD11b⁺GR-1⁺ MDSCs on Day15 after B16.F10 inoculation of PBS-treated C57BL/6 mice ($n = 5$), PBS-treated αSMA$^{cre}$Atg5$^{fl/fl}$ mice ($n = 6$), anti-PD-1/anti-CTLA-4-treated C57BL/6 mice ($n = 5$), anti-PD-1/anti-CTLA-4-treated αSMA$^{cre}$Atg5$^{fl/fl}$ mice ($n = 6$). Data are shown as mean ± SD. Representative data from three independent experiments are shown; 4−9 mice/group were used in each experiment. One-way ANOVA with Tukey's multiple comparison tests (**b**, **c**); two-tailed Log-rank test (**b**). $n =$ biologically independent mouse samples. $P$ values are as indicated in the respective graph. Source data are provided as a Source Data file.

characterized by an ample presence of autophagosome structures, while specific ablation of the autophagic pathway leads to break down of synapse formation with Treg cells, accompanied by tumour regression and enhanced efficacy to ICI dual immunotherapy.

Foxp3[+] T regulatory (Treg) cells are robust suppressors of anti-tumour immune responses, promoting tumour development[41,42]. Increased Treg cell frequencies in the tumour stroma are associated with unfavourable prognosis and immunotherapy resistance in several cancer types[42,43]. Various attempts have been explored to therapeutically target the potent immunosuppressive function of Treg cells with the ultimate goal to empower the anti-tumour immunity. For example, interference with Foxp3-TSDR demethylation in Treg cells caused loss of Foxp3 expression (ex-Tregs) and acquisition of an inflammatory function leading to tumour rejection[44]. In a similar fashion, several pathways have been described to facilitate the induction of "fragile" Treg cells, which retain Foxp3 expression but express high levels of pro-inflammatory cytokines such as IFN-γ, potentiating the anti-tumour immunity[45–47]. IFN-γ was also a fingerprint cytokine expressed by Treg cells in the peripheral blood of melanoma patients treated with ICI immunotherapy and developed immune-related adverse events, indicating a disturbance of peripheral tolerance[48]. Despite the existing knowledge on cell-intrinsic mechanisms that can intervene with Treg cell suppressive function, this information has not been transformed into clinical practice. One possible scenario is that interfering with Treg stability may fuel the development of autoimmunity. Thus, it is crucial to delineate TME-specific mechanisms that underlie Treg cell suppressive function. In this line, we identify a cell-extrinsic mechanism in which α-SMA[+] CAFs form immunological synapses with Treg cells in the TME, promoting their activation and proliferation. Several studies have proposed a potential crosstalk between CAFs and Tregs. For example, CD4[+]CD25[+]Foxp3[+] Tregs were found in close proximity to α-SMA[+] stromal cells in mouse and human breast cancer[21,49], or to GP38[+] fibroblasts in a lung adenocarcinoma model[27]. CAFs were proposed to chemoattract Tregs through secretion of CCL17 and CCL22 in oral squamous cell carcinoma patients[23] or CCL5[50], CXCL12[21] secretion in the tumour site of patients with breast cancer. Our findings extend this notion by revealing the formation of synapses between α-SMA[+] CAFs and Treg cells in the TME while demonstrating autophagy and PD-1/PD-L1 axis to constitute essential components of α-SMA[+] CAF-Treg cell interactions. In this context, we also show that partial depletion of α-SMA[+] CAFs in the poorly immunogenic B16.F10 melanoma promotes tumour regression through the development of anti-tumour immunity as demonstrated with the increased frequencies of activated PD1[+] CD4[+] T effector cells, IFNγ-producing CD8[+] cytotoxic T cells and accompanied by decreased Treg cell frequencies in the TME. In addition, we observed the up-regulation of MHC class II expression in CD11c[+] and CD11b[+]GR-1[+] myeloid cells, indicating that the absence of α-SMA[+] CAFs may exert additional immunogenic effects in immune cell populations infiltrating the TME. Although several mechanisms may account for the upregulation of MHC class II expression ranging from Treg cell-dependent (i.e., lack of IL-10 secretion[47]) to Treg cell-independent (i.e., increased IFN-γ regulates the expression of the class II trans-activator CIITA[47]), these findings are consistent with the development of anti-tumour immunity. Despite the fact that a large body of literature, including our study, proposes a pro-tumorigenic role of α-SMA[+] CAFs, an anti-tumorigenic role has also been reported. Specifically, in pancreatic ductal adenocarcinoma (PDAC)[27] and in metastatic colorectal cancer[51], depletion of α-SMA[+] CAFs led to more aggressive tumours and increased frequencies of Treg cells in the TME. Several reasons may account for the different outcomes reported in the aforementioned studies, such as the immunogenicity of the tumour, the extent of fibrosis in the specific mouse model and how these features were affected by α-SMA[+] CAF depletion and also by the time period and extent of CAF depletion. More studies in experimental models are required in order to shed light on the potentially divergent mechanisms that underlie the decision on a pro- or anti-tumorigenic role of α-SMA[+] CAFs.

Of interest, α-SMA[+] CAFs express a tolerogenic phenotype in the antigen-presentation machinery, which is known to favour Treg cell proliferation. Importantly, we demonstrate that α-SMA[+] CAFs express the co-inhibitory molecules PD-L1 and PD-L2, which have extensively been implicated in the induction of Treg cells[52,53]. This finding may possess major therapeutic implications since the PD-1/PD-L1 axis is a major target of ICI immunotherapy and thus may also contribute to immunotherapy resistance which develops in a large proportion of patients. Although anti-PD-L1 treatment has yielded debatable results on whether it successfully decreases the tumour burden[54–56], in support of the above, anti-PD-L1 treatment of melanoma-bearing animals did not significantly promote tumour regression, while the reciprocal communication between CAFs and Treg cells remained unaltered in the TME, which may be attributed to anti-PD-L1 failure to disrupt the synapse formation of α-SMA[+] CAFs with Tregs. Supporting this hypothesis, PD-1[−/−] Tregs showed reduced accumulation in α-SMA[+] CAF-rich areas, denoting that the PD-L1/PD-1 axis is essential for the engagement of the two cell types. Further studies addressing the molecular circumstances dictating binding between α-SMA[+] CAFs and Treg cells via PD-L1/PD-1 shall shed light on mechanisms governing cancer pathogenesis and immunotherapy resistance.

TCR engagement is indispensable for Treg cell activation proliferation and function[57,58]. We show that α-SMA[+] CAFs possess the ability to present antigens in the context of MHC II and to promote the activation and proliferation of antigen-specific Foxp3[+] Treg cells. These findings are in accordance with various studies that describe the existence of CAF subpopulations in the TME with the ability to present antigens, termed antigen-presenting CAFs (apCAFs), which may express α-SMA, and contribute to chemoattraction[21] and activation[21,26] of Foxp3[+] Treg cells. Whether CAFs bear the unique ability to activate Treg cells but not CD4[+] T effector cells (Teffs) of CD8[+] T cells remains as an open question. Our data show that CD4[+] Teff cells also accumulate in α-SMA[+] CAF-rich areas while CD8[+] T cells were equally distributed in α-SMA[+] CAF-rich and -poor areas in the TME. Since α-SMA[+] CAFs express low levels of MHC II molecules, we would expect to favour Treg cell activation as supported by our study. Furthermore, several CAF subsets may co-localize with α-SMA[+] CAFs to instruct activation of CD4[+] Teff and/or CD8[+] T cells or even conversion to Foxp3[+] Treg cells in support to data presented in a model of pancreatic cancer[26]. In line with this, Kerdidani et al.[59] showed that apCAFs (which do not express α-SMA) facilitate the expansion of CD4[+] T effector cells in the TME and promote tumour regression in a mouse model of non-small cell lung cancer[59]. In this regard, the question that has not been addressed in the CAF field so far is the mechanisms via which CAFs can process tumour antigens in order to explain their antigen-presenting properties. To this end, our findings demonstrate that α-SMA[+] CAFs can phagocytose tumour antigens, which is consistent with literature showing that dermal fibroblasts can acquire melanosomes and also show the ability to process model antigens in vitro[60]. Of interest, α-SMA[+] CAFs showed an abundance of autophagosome presence in their cytoplasm, suggesting increased operation of autophagy. This is also supported by a meta-analysis of published scRNA-seq data which showed enhanced expression of the autophagy pathway on α-SMA-expressing CAF clusters. On a functional note, autophagy ablation on α-SMA[+] CAFs promotes tumour regression and unleashes the Treg cells from the CAF vicinity, indicating that tumour antigen-processing may be facilitated through autophagy in α-SMA[+] CAFs. Indeed, autophagy has an established role in antigen processing and presentation in diverse inflammatory settings, including cancer. Although the signals that activate CAF autophagy in the TME, as well as the type of autophagy engagement in CAFs, remain unexplored, its targeting leads to a transcriptomic and proteomic re-programming consistent

with an inflammatory phenotype and thus autophagy may be considered as a potential switch from pro- to anti-tumorigenic properties of CAFs. Importantly, conditional ablation of autophagy in α-SMA⁺ CAFs demonstrates a synergistic effect with ICI immunotherapy, suggesting that CAF autophagy may constitute a target that should be explored therapeutically. Since aCTLA-4 and aPD-1 engage different mechanisms to induce anti-tumour immunity[61], such as expansion of CD4⁺ T cells and increase of T cell mobility into "cold" tumours for the former, expansion of CD8⁺ T cells and induction of differentiated T cell function for the latter, we hypothesized that several mechanisms may operate in concert upon aCTLA-4/aPD-1 treatment in mice with autophagy-ablated α-SMA⁺ CAFs to promote anti-tumour immunity. The inflammatory re-programming of CAFs upon autophagy ablation, as demonstrated by our transcriptomic and proteomic data, may also assist in immunotherapy efficacy, however, the precise mechanism(s) through which deletion of autophagy in α-SMA⁺ CAFs synergizes with immune checkpoint inhibitors remain to be investigated.

To conclude, an in-depth understanding of the immunosuppressive mechanisms operating in the TME will provide fundamental opportunities for therapeutic interventions in cancer. Therefore, considering the unique properties of CAFs in promoting cancer development and being resistant to radiotherapy and chemotherapy, combined with the current findings on their role in promoting Treg cell activation and proliferation, it is apparent that CAF-targeted therapies may pave the way to the development of robust and durable anti-tumour immune responses.

## Methods

### Animals
C57BL/6 mice were purchased from Jackson Laboratory (stock #000664). αSMA-tk mice (C57BL/6 background) and α-SMA^cre mice (C57BL/6 background) were kindly provided by Dr. Kalluri (Department of Cancer Biology, Metastasis Research Center, University of Texas MD Anderson Cancer Center, Houston, TX, USA)[24] and α-SMA-RFP mice (C57BL/6 background) were purchased from Jackson Laboratory (stock #031159). Atg5^fl/fl mice (C57BL/6 background) were kindly provided by Dr. Mizushima (RIKEN BioResource Center)[62]. αSMA-tk;RFP mice were generated by crossing αSMA-tk mice with α-SMA-RFP mice and α-SMA^creAtg5^fl/fl mice were generated by crossing α-SMA^cre mice with Atg5^fl/fl mice. Foxp3gfp.KI mice (C57BL/6 background) were kindly provided by Dr. Rudensky (Department of Immunology, Memorial Sloan-Kettering Cancer Center, New York, New York, USA)[63]. OTII mice (C57BL/6 background) were kindly provided by Dr. Xanthou (Center for Basic Research, BRFAA, Athens, Greece). RAG⁻/⁻/OTII mice (C57BL/6 background) were kindly provided by Dr. Tsoumakidou (Institute of Bioinnovation, Biomedical Sciences Research Center "Alexander Fleming", Vari, Greece)[59]. PD-1⁻/⁻ mice (C57BL/6 background) was kindly provided by Dr. Zhang (Department of Orthopedic Surgery, University of Chicago, Chicago, IL, USA)[64]. All mice were maintained in the specific pathogen-free (SPF) animal facility of the Biomedical Research Foundation of the Academy of Athens (BRFAA); the mice were kept in a 12-h light–dark cycle at a room temperature of 20–24 °C and a humidity range of 45–65%. Experimental and control animals were co-housed. The protocols used for animal experimentation were in accordance with institutional guidelines and approved by the Welfare Institutional Committee of Protocol Evaluation together with the Directorate of Agriculture and Veterinary Policy, Region of Attika, Greece (protocol No. 530216/22 July 2020). In all experiments, sex-matched mice aged between 8 and 12 weeks were used, and at the experiment endpoint, mice were euthanized by cervical dislocation.

### Cell lines
The B16.F10 mouse melanoma, Lewis Lung Carcinoma (LLC), and MB49 mouse bladder carcinoma cell lines were kindly provided by Dr. Eliopoulos (School of Medicine, University of Athens, Greece). All cell lines were negative for mycoplasma, as confirmed by PCR. B16.F10 and LLC cells were maintained in RPMI Medium (Gibco) supplemented with 10% foetal bovine serum (FBS, StemCell Technologies Inc.), 0.1% β-mercaptoethanol (Gibco) and 1% of a mix of Penicillin/Streptomycin (P/S, Gibco). MB49 cells were maintained in DMEM Medium (Gibco) supplemented with 10% FBS, 0.1% β-mercaptoethanol and 1% of a mix of P/S.

### In vivo protocols
**Solid tumour induction.** Induction of solid tumours was performed as previously described[46,65]. Briefly, C57BL/6, αSMA-tk, α-SMA-RFP, αSMA-tk;RFP, α-SMA^creAtg5^fl/fl and PD-1⁻/⁻ mice were inoculated subcutaneously (s.c.) on the back with $3 \times 10^5$ B16.F10 cells or $3 \times 10^5$ LLC cells or $6 \times 10^5$ MB49 cells. Tumour growth was monitored every day from Day 9 to Day 15 by measurement of two perpendicular diameters (*d*) of the tumour by caliper; tumour volume was calculated using the equation $(d1*d2*d2)/2$. Mice were euthanized when tumours grew larger than 2000 mm³. At the endpoint of each experiment, tumour weight was determined.

**Fibroblast depletion experiments.** For fibroblast depletion experiments, B16.F10 melanoma-bearing αSMA-tk, αSMA-tk;RFP mice and C57BL/6 mice received intraperitoneal (i.p.) injections with 12.5 mg/kg of body weight of ganciclovir (GCV, Cymevene®, Roche) every 48 h, diluted in 100 μl phosphate buffer saline (PBS). As control, B16.F10 melanoma-bearing αSMA-tk, and αSMA-tk;RFP mice received i.p. injections with 100 μl PBS every 48 h. GCV/PBS injections were initiated on the day mice were inoculated with B16.F10 melanoma cells. Littermates of the same genotype were randomly allocated to experimental groups.

**Immunotherapy experiments.** α-SMA-RFP mice received i.p. injections with anti-PD-L1 (clone MIH5, 200 μg per mouse) in 100 μl PBS, every 3 days following B16.F10 inoculation. As a control, B16.F10 melanoma-bearing α-SMA-RFP mice received i.p. injections with 100 μl PBS every 3 days. C57BL/6 mice received i.p. injections with anti-CTLA-4 (clone 4F10, 200 μg per mouse) in 100 μl PBS, every 3 days following B16.F10 inoculation. As a control, B16.F10 melanoma-bearing C57BL/6 mice received i.p. injections with 100 μl PBS every 3 days. α-SMA^creAtg5^fl/fl mice and Atg5^fl/fl or C57BL/6 mice received i.p. injections with anti-PD-1 (clone RMP1–14, 200 μg per mouse) in 100 μl PBS, every 3 days following B16.F10 inoculation. As a control, B16.F10 melanoma-bearing α-SMA^creAtg5^fl/fl, Atg5^fl/fl, or C57BL/6 mice received i.p. injections with 100 μl PBS every 3 days. For the application of dual immunotherapy, α-SMA^creAtg5^fl/fl mice and Atg5^fl/fl or C57BL/6 mice received i.p. injections with a combination of anti-CTLA-4 (clone 4F10, 200 μg per mouse) and anti-PD-1 (clone RMP1–14, 200 μg per mouse) in 200 μl PBS, every 3 days following B16.F10 inoculation. As control, B16.F10 melanoma-bearing α-SMA^creAtg5^fl/fl, Atg5^fl/fl, or C57BL/6 mice received i.p. injections with 200 μl PBS every 3 days. Littermates of the same genotype were randomly allocated to experimental groups.

**Mouse genotyping.** Mice were genotyped according to previously published protocols regarding the αSMA-tk mice and αSMAcre mice from Jackson Laboratories (Bar Harbor, ME, USA) and the Atg5fl/fl mice from RIKEN BioResource Center (Koyadai, Tsukuba-shi, Ibaraki, Japan). For the specific primers used, please see Supplementary Table 1.

**Cell isolation from tumours and lymphoid organs for flow cytometry.** Single-cell suspensions from mouse lymph nodes and spleens were generated by passing them through a 40 μm cell strainer (BD Falcon). For tumour cell analysis, melanoma tissues were excised and cut into the smallest possible fragments by using an ophthalmic scissor. The minced tissues were incubated for 45 min at 37 °C in RPMI medium containing DNase I (0.25 mg/ml, Sigma) and collagenase D

(1 mg/ml, Roche). For analysis of tumour infiltrating lymphocytes (TILs), cell suspensions were prepared by passing through a 40 µm cell strainer (BD Falcon). For analysis of CAFs, cell suspensions were prepared by passing through a 100 µm cell strainer (BD Falcon).

**Fibroblast isolation.** For the isolation of CAFs from dissociated tumours, the mouse Tumour-Associated Fibroblast Isolation Kit was used (Miltenyi Biotec); isolation was performed according to manufacturer instructions applying a two-step strategy. Briefly, samples were first pre-enriched by removing non-target cells and isolation using the marker CD90.2 was followed. For pre-enrichment, LD Columns on a MidiMACS™ Separator (Miltenyi Biotec) were used and for the subsequent CAF isolation, MS Columns on a MiniMACS™ Separator (Miltenyi Biotec) were used. Isolated CAFs were cultured in α-MEM medium supplemented with 10% FBS, β-mercaptoethanol and a mix of P/S.

**Flow cytometry and cell sorting.** For staining of extracellular markers, cell suspensions were incubated in PBS−5% FBS with antibodies for 20 min at 4 °C. The following antibodies for extracellular markers were purchased by Biolegend and used at a dilution of 1:200: CD45 (clone 30-F11, cat. number 103132), CD4 (clone GK1.5, cat. number 100402), CD4 (clone RM4-4, cat. number 116008), CD4 (clone RM4.4, cat. number 116006), CD8 (clone 53-6.7, cat. number 100722), PD-1 (clone 29F.1A12, cat. number 135218), CD31 (clone MEC13.3, cat. number 102522), CD90.2 (Thy1.2) (clone 30-H12, cat. number 105328), CD90.2 (Thy1.2) (clone 53-2.1, cat. number 140304), PD-L1 (clone 10F.9G2, cat. number 124315), PD-L1 (clone 10F.9G2, cat. number 124308), CD11c (clone N418, cat. number 117318), CD11b (clone M1/70, cat. number 101206), Ly-6G/Ly-6C (Gr-1) (clone RB8-8C5, cat. number 108408), I-Ab (clone AF6-120.1, cat. number 116406), CD80 (clone 16-10A1, cat. number 104707), CD86 (clone PO3, cat. number 105109), CD25 (clone PC61, cat. number 102034), GITR (clone DTA-1, cat. number 126308), TCR Vβ5.1, 5.2 (clone MR9-4, cat. number 139507), CD44 (clone IM7, cat. number 103032), F4/80 (clone BM8, cat. number 123110), CD69 (clone H1.2F3, cat. number 104507), CD146 (clone ME-9F1, cat. number 134710), CD326 (Ep-CAM) (clone G8.8, cat. number 118216), CD54 (clone YN1/1.7.4, cat. number 116105). In addition, the following antibodies for extracellular markers were purchased by BD Biosciences and used at a dilution of 1:200: PD-L1 (clone MIH5, cat. number 564716), H-2Kb (clone AF6-88.5, cat. number 553569), CD48 (clone HM48-1, cat. number 740353). The following antibodies for extracellular markers were purchased by ThermoFisher Scientific and used at a dilution of 1:200: TCR Vα2 (clone B20.1, cat. number 12-5812-82), PD-L2 (clone 122, cat. number 11-9972-82). For Foxp3 (1:50, Biolegend, clone 150D, cat. number 320012), Foxp3 (1:50, Biolegend, clone MF014, cat. number 126410) and Ki-67 (1:50, Biolegend, clone 16A8, cat. number 652425) intracellular staining, cells were stained for the extracellular markers and then fixed and stained using the Foxp3/Transcription Factor Staining Buffer Set (eBioscience) according to manufacturer instructions. For α-SMA (1:100, Abcam, clone 1A4, cat. number ab8211) staining, cells were stained for the extracellular markers and then fixed and stained using the Intracellular (IC) Fixation and Permeabilization Buffer (eBioscience) according to manufacturer instructions. For IFN-γ (1:50, Biolegend, clone XMG1.2, cat. number 505808) intracellular staining, cells were incubated with 50 ng/ml PMA (Sigma-Aldrich), 2 µg/ml Ionomycin (Sigma-Aldrich) and Brefeldin (1/1000) (BD) for 6 h at 37 °C and 5% $CO_2$, stained for extracellular markers, fixed and stained using the Foxp3/Transcription Factor Staining Buffer Set (eBioscience) according to manufacturer instructions. CD4⁺CD25⁺GITR⁺Vα2⁺Vβ5.1⁺ OTII Tregs, CD4⁺CD25⁻GITR⁻Vα2⁺Vβ5.1⁺ OTII T cells, CD45⁻CD31⁻RFP⁺ CAFs were sorted on a FACS ARIA III (BD Biosciences), utilizing a BD FACS Diva Software v 6.1.3. Cell purity was above 95%. Flow cytometry data were analysed with FlowJo (v10 and v10.7.2) software (Tree Star).

**Mouse histological analysis and tissue immunofluorescence.** Lungs, intestines and melanoma tumours were isolated either from naïve or B16.F10 tumour-bearing αSMA-tk mice, α-SMA^cre Atg5^fl/fl mice and Atg5^fl/fl mice. Tissues were fixed in 10% formalin solution overnight, transferred to 70% EtOH and embedded in paraffin. Formalin-fixed, paraffin-embedded sections were stained with Hematoxylin and Eosin (H&E), using standard histology procedures. For immunofluorescence staining of murine tumour tissues, harvested B16.F10, Lewis Lung Carcinoma and MB49 tumour samples were fixed overnight in 4% PFA/PBS, washed with PBS and immersed in 30% sucrose/PBS. Then, tumours were embedded in O.C.T. medium (VWR Chemicals) and frozen. 5–10 µm cryosections were blocked with 5% bovine serum albumin (BSA) and 0.3 M glycine in PBS and immunostained using standard protocols. Primary antibodies used: anti-VCAM-1 (1:500, clone M/K-2, Abcam, cat. number ab19569), anti-Foxp3 (1:100, Novus Biologicals, cat. number NB100-39002), anti-Foxp3 (1:100, clone FJK-16s, eBioscience, 14-5773-82), FITC anti-CD4 (1:100, clone GK1.5, Biolegend, cat. number 100406), FITC anti-CD8α (1:100, clone 53-6.7, Biolegend, cat. number 100706), FITC anti-α-SMA (1:100, Abcam, clone 1A4, cat. number ab8211), AF488 anti-CD3 (1:20, Biolegend, clone 17A2, cat. number 100212), BV421 anti-I-A/I-E (1:100, M5/114.15.2, Biolegend, 107631). Foxp3 was detected by staining with Alexa Fluor 647 (A647) anti-rabbit IgG (1:200; A21245; Invitrogen); VCAM was detected by staining with Alexa Fluor 555 (A555) anti-rat IgG (1:200, A11006, Invitrogen). DAPI (Molecular Probes) was used to stain cell nuclei, and sections were mounted using Mowiol (Sigma). Images were acquired using an inverted confocal live cell imaging system Leica SP5, utilizing a Leica Application Suite X (LAS X) as a software platform (Leica Microsystems). Quantifications were performed using ImageJ software analysis. All measurements were performed in the stromal area, not considering any positive signal in tumour-adjacent areas.

**Quantification.** Quantification of Treg cells was performed in the stromal compartment only by using ImageJ software analysis. Tregs in at least 5–10 representative fields at 20x magnification per tumour were counted manually and divided by the pixels of the area of the section considered in a blind manner. Analysis of the area occupied by α-SMA⁺ CAFs was conducted in ImageJ software, using an in-house developed pipeline. To distinguish CAF-rich vs CAF-poor areas of the stroma, the median α-SMA⁺ CAF area was used as a threshold in every experiment.

**Autophagic flux assay.** CAFs were magnetically isolated from Day15 B16.F10 melanomas of C57BL/6 mice or α-SMA^cre Atg5^fl/fl mice using the mouse Tumour-Associated Fibroblast Isolation Kit as already described (Miltenyi Biotec). $5 \times 10^4$ cells CAFs were seeded on poly-L-Lysine coated glass coverslips, treated with the lysosomal inhibitor $NH_4Cl$ (20 mM; Sigma-Aldrich) or $H_2O$ as vehicle control, and incubated overnight in complete medium (α-MEM, 10% FBS; 37 °C, 5% $CO_2$). Next day, cells were washed with PBS (10 min, RT), fixed and stained for autophagy markers according to the autophagy immunofluorescence staining protocol described below.

**Immunofluorescence cell staining.** For autophagic flux assay, $NH_4Cl$-treated CAFs or vehicle controls isolated from Day15 B16.F10 melanomas of C57BL/6 mice or α-SMA^cre Atg5^fl/fl mice. For Foxp3 staining of T cells, CD4⁺Foxp3⁻ T effector cells (Teffs), CD4⁺Foxp3⁺ T regulatory cells (Tregs) and CD8⁺ cytotoxic T cells were isolated by FACS sorting from total lymph nodes and spleens of Foxp3gfp.KI mice. Cells were fixed with 4% paraformaldehyde (PFA) in PBS for 15 min at room temperature (RT), followed by post-fixation with ice-cold methanol (−20 °C; Sigma-Aldrich) for 10 min, RT and washed with PBS.

**Autophagic flux assay.** Cells were blocked with Permeabilization Buffer I (PSI) (0.1% saponin, 2% BSA in PBS; 15 min, RT) and stained with

the following primary antibodies: Lamp-1 (rat, 1:400, clone 1D4B, Santa Cruz Biotechnology, cat. number sc-19992), p62 MBL (rabbit, 1:500, MBL, PM045) and LC3 (mouse, 1:20, clone 5F10, NanoTools, cat. number 0231-100/LC3-5F10) (in PSI Buffer; 1 h, RT), and secondary antibodies: Alexa Fluor 555 anti-mouse IgG (1:500, Invitrogen, cat. number A21425), Alexa Fluor 647 anti-rabbit IgG (1:200, Invitrogen, cat. number A21245), and Alexa Fluor 488 anti-rat IgG (1:250, Invitrogen, cat. number A11006) (in PSI Buffer; 45 min, RT). For visualization of the nuclei, cells were stained with DAPI (1:100, 3 min, Sigma-Aldrich). Images were obtained using an inverted confocal live cell imaging system Leica SP5. The numbers of LC3 spots/cell, p62 spots/cell, Lamp-1 spots/cell were calculated using a macro developed in Fiji software v 2.3.0 (SciJava), as previously described.

**Foxp3 staining in T cells.** Cells were treated with ice-cold methanol (5′) to remove endogenous fluorescence and subsequently blocked with Permeabilization Buffer II (PSII) (0.3% Triton, 2% BSA in PBS; 30 min, RT). Then, cells were stained with anti-Foxp3 (1:100, Novus Biologicals, cat. number NB100-39002) (in PS buffer, 1 h, RT) and secondary antibody Alexa Fluor 647 (A647) anti-rabbit IgG (1:200, Invitrogen, cat. number A21245) (in PS buffer, 1 h, RT). For visualization of the nuclei, cells were stained with DAPI (1:100, 3 min; Sigma-Aldrich). Images were obtained using an inverted confocal live cell imaging system Leica SP5.

**Visualization of CAF-Treg immune synapses.** Cells were permeabilized with 0.1% Triton in PBS and blocked with Image-iT FX Signal Enhancer (30′, RT). Then, cells were stained with anti-mouse CD16/CD32 (1:20, clone 93, eBioscience, cat. number 14-0161-86), Alexa Fluor 647 Fab-fragment of anti-TCRβ (1:50, clone, H57-597, eBioscience, cat. number 14-5961-82) and FITC anti-PD-1 (1:50, clone J43, eBioscience, cat. number 11-9985-85). Anti-mouse PD-1 FITC was further targeted with Alexa Fluor 488 anti-fluorescein (1:100, 200-542-037, Jackson Immuno Research).

## Human tumour samples

**Study population.** The current study included 15 surgically resected stage II and III colorectal carcinoma cases and three melanoma cases prior to any treatment. Twenty-four tissue microarray (TMA) paraffin blocks were constructed by selecting 2 mm cores from the centre and the invasive margin of tumour areas. The study was approved by the ethics committee/institutional review board of Attikon University Hospital, Haidari, Athens, Greece (protocol # 2nd Department of Pathology, EBΔ444/17-12-2010) and conducted in accordance with the 1964 Declaration of Helsinki and its later amendments. Written informed consent was obtained to use sociodemographic data from all patients and for the use of their tumour samples as tumour microarrays (TMA).

**Immunohistochemistry.** Double immunohistochemistry was performed on 2-μm-thick deparaffinized sections by using primary antibodies against Foxp3 (rabbit monoclonal, clone SP97 [Zytomed], 1:50 dilution, 30 min incubation) and α-SMA (mouse monoclonal, clone 1A4 [Dako], 1:200 dilution, 30 min incubation), respectively. Immunostaining was performed using the Dako Autostainer Link 48 devise, with the EnvisionTM Flex HRP High pH kit (K8000, Dako). Deparaffinized sections were immersed in EnVision FLEX Target Retrieval Solution, high pH (DM828, Dako) and boiled in the PT LINK pre-treatment module (Dako) and subsequently cooled at room temperature for 20 min. Endogenous peroxidase activity was blocked by means of the EnVision FLEX peroxidase-blocking reagent (SM801, Dako). EnVision Flex Substrate buffer (SM803, Dako) with DAB chromogen (DM827, Dako) or with magenta chromogen (DM857, Dako) were used for visualization of Foxp3 and SMA, respectively. All sections were lightly counterstained with EnVision FLEX Hematoxylin (SM806, Dako) for 45 s prior to mounting.

**Quantification.** Quantification of Treg cells was performed in the epithelial and stromal compartments separately by using ImageJ software analysis. Tregs in at least 5–10 representative fields at ×20 magnification per tumour were counted manually and divided by the pixels of the area of the section considered in a blind manner.

## Bulk RNA-seq data analysis

**RNA-sequencing library preparation.** CAFs were magnetically isolated from fully grown (Day15) melanomas of α-SMA$^{cre}$Atg5$^{fl/fl}$ and Atg5$^{fl/fl}$ mice using the mouse Tumour-Associated Fibroblast Isolation Kit as already described (Miltenyi Biotec). RNA was extracted with the Macherey-Nagel NucleoSpin® RNA kit, according to manufacturer instructions. Each RNA sample represented one biological replicate. NGS libraries were generated using 300 ng total RNA as input on average with the QuantSeq 3′ mRNA-Seq Library Prep Kit FWD for Illumina kit from Lexogen according to manufacturer protocol, using 16 or 19 cycles of amplification. Libraries were sequenced on Illumina Nextseq 500 on 1 × 75 High flowcell.

**Preprocessing.** Raw single-end fastq files from six mice were trimmed with *trimmomatic (v0.39)*, deduplicated with *clumpify (BBMap v38.95)* and aligned using reference genome *GRCm38 (mm10)* with *hisat2 (v2.2.1)*. In order to assess read quality, we ran trimmomatic with different CROP flags in the range of [30–75] with a step of 5, basing our selection on the average overall alignment rate from hisat2. CROP:60 was selected, which achieved an average overall alignment rate of 86.82%, while hisat2 was run with -k 2 and --score-min L,0,−5. Finally, counting was performed with *htseq-count (v1.99.2)* with -s yes and the appropriate *GTF (mm10)* file.

**Differential expression analysis.** Whole analysis was performed with the *R* library edgeR (v3.36.0). Genes with zero expression on all samples were discarded, while further filtering out was performed with filterByExpr using min.count = 10 and min.prop = 0.4. Normalization was carried out with the functions calcNormFactors, estimateDisp and cpm. Differential expression analysis was executed with the functions glmFit and glmLRT, reporting genes with adjusted *p*-value (FDR) < 0.05 and absolute log2 fold change > 1.5 as differentially expressed (DEGs). Heatmaps were generated with the R library ComplexHeatmap.

**Enrichment analysis.** *Pathway* (Reactome and KEGG) and *gene ontology* (GO) enrichment analyses were carried out with the *R* library clusterProfiler. Genes were ranked using the signed statistic from the differential expression analysis. Here, two different sets of ranking systems were used; ranking genes from most downregulated to most upregulated with respect to the control group and using the absolute value of the statistic, resulting in deregulated gene-lists where the most significant genes lie to the left-most side of the produced GSEA plots.

## Single-cell RNA-seq meta-analysis

**Datasets.** Single-cell datasets were downloaded from Davidson et al.[32], Elyada et al.[16] and Grauel et al.[18] according to authors' instructions. CAF-related cell populations were defined by gene expression levels as described in Grauel et al.[18]. The R library Seurat, along with the respective functions, was used throughout the analysis of these datasets.

**Data integration.** Grauel et al.[18] single-cell dataset was used as a reference to map cell populations on a common feature space using the functions FindTransferAnchors and MapQuery. Cell populations and CAF identification were described using the gene signature: *Col1a1, Col3a1, Thy1, Pdpn, Fap, Epcam, Cd74* and *Lyz2*.

**Differential expression analysis.** *Acta2* gene-expression distribution was used to assign CAF cells to three groups based on their expression

levels (quantiles): Low (0–40%), Average (40–60%) and High (60–100%). DE analysis was carried out with the function FindMarkers, for 2 settings: Low vs. Average and Low vs. High. A subsequent DE analysis was carried out where the gene-expression distribution of *Acta2* was grouped between cells with zero expression and cells expressing even the slightest quantity of RNA, denoted as – and +, respectively.

**Expression visualization.** Expression of gene groups of interest was visualized through Seurat's FeaturePlot function; Gene signature of CAFs was augmented with more genes, including *Acta2, Col1a1, Col3a1, Thy1, Pdpn, Fap, Epcam, Ptprc, Cd74, Mcam, Pecam1* and *Cspg4* while their expression level was demonstrated across all CAF cluster groups. Co-expression plots of *Acta2* levels versus *Cd74, H2-Aa* and *H2-Ab1* levels were reported also through FeaturePlot function with flag blend set to "True". Expression of key genes contributing to pathways of interest, i.e., MHC class II (*Cd74, H2-Aa, H2-DMa* and *H2-DMb1*) and Autophagy (*Lamp2, Wipi1, Map1lc3a, Gabarap*l and *Tmem59*) was also visualized through ggplot's geom_violin function. Further statistical comparisons of distributions of specific pathway gene's expression in *Acta2*+ and *Acta2*– cells were evaluated through the Wilcoxon test (significance labels: "****"0.001, "***"0.01, "**"0.05 and "NS" = non-significant).

**Phagocytosis assay.** For phagocytosis of tumour cells, isolated CAFs were stained with the CYTO-ID® Red long-term cell tracer kit (Enzo) according to manufacturer instructions and seeded in 12-well plates at a density of 100,000 cells/3.5 cm². The next day, B16.F10 cells were stained with CMFDA Cell Tracker (1 μM, Thermo Fisher Scientific) according to manufacturer instructions and co-cultured with CAFs at a ratio of 1:1. After incubation for 5 h at 37 °C, 5% $CO_2$, the phagocytosis of melanoma cells was assessed with flow cytometry. For phagocytosis of tumour cells dying via non-immunogenic cell death, B16.F10 cells were treated for 24 h with cisplatin (500 μM, EMD Millipore Corp.). For the reduction of receptor–ligand binding kinetics, which resulted in phagocytosis inhibition, a duplicated experiment was conducted in parallel at 4 °C. Apoptotic B16.F10 cells were stained with 7-Amino-actinomycin D (7-AAD, Biolegend) to determine viability. Flow cytometry data were analysed with FlowJo (v10 and v10.7.2) software (Tree Star). % phagocytosis was calculated as the percentage of the total number of CAFs containing cancer cells.

**In vitro ovalbumin uptake and processing.** For analysis of antigen processing, CAFs were seeded in 24-well plates at a density of 50,000 cells/1.9 cm² and pulsed with DQ-ovalbumin (DQ-OVA, 100 μg/ml, Invitrogen) for 1 h. Cells were then washed with ice-cold PBS 5% FBS and cultured in pre-warmed complete medium (α-MEM medium supplemented with 10% FBS, β-mercaptoethanol and a mix of P/S) for 1 h. Processing of DQ-OVA was determined with flow cytometry. Also, CAFs were incubated in the presence of the lysosomal inhibitor ammonium chloride (NH₄Cl, 15 mM, Sigma) overnight and on the next day, they were pulsed with DQ-OVA.

**In vitro co-culture experiments for antigen presentation assessment.** For antigen-specific CAF-Treg interactions, we performed co-culture experiments. Highly pure (>95%) CD4+CD25+GITR+Vα2+Vβ5.1+ Tregs were isolated by FACS sorting from total lymph nodes and spleens of RAG⁻/⁻/OTII naive mice. Isolated CAFs were cultured in 1:1 ratio with sorted Tregs in the presence of OVA₃₂₃₋₃₃₉ peptide (20 μg/ml, Caslo ApS) and IL-2 (5000 U/ml, Peprotech). As a positive control, Tregs were incubated in the presence of IL-2 and beads coated with monoclonal antibody (mAb) to CD3 plus mAb to CD28, at a ratio of 1 bead per 1 cell (Invitrogen), and as negative control, CAFs were co-cultured with Tregs in the presence of IL-2 without OVA₃₂₃₋₃₃₉ peptide.

For autophagy-mediated antigen-specific CAF-Treg interactions, highly pure (>95%) CD4+CD25⁻GITR⁻Vα2+Vβ5.1+ T effector cells (Teffs) and CD4+CD25+GITR+Vα2+Vβ5.1+ T regulatory cells (Tregs) were isolated by FACS sorting from total lymph nodes and spleens of OTII naive mice. Isolated CAFs from Day15 B16.F10 melanomas of C57BL/6 mice or α-SMAᶜʳᵉAtg5ᶠˡ/ᶠˡ mice were cultured in a 1:1 ratio with sorted Teffs or Tregs in the presence of OVA protein (50 μg/ml). As a positive control, FACS-sorted dendritic cells were co-cultured in 1:1 ratio with Teffs or Tregs in the presence of OVA protein and as a negative control, CAFs were co-cultured in a 1:1 ratio with Teffs or Tregs without OVA protein. IL-2 (5000 U/ml, Peprotech) was added in all wells containing Treg cells. In all settings, T cell activation and proliferation were analysed 3 days later with flow cytometry.

**Co-culture of Tregs with CAFs for time-lapse video microscopy.** For co-culture experiments for time-lapse video microscopy, 5–8000 CAFs were seeded in eight-well chamber slides (Ibidi) and allowed to attach overnight in a complete medium. The next day, highly pure (>95%) CD4+CD25+GITR+Vα2+Vβ5.1+ Tregs were isolated by FACS sorting from total lymph nodes and spleens of RAG⁻/⁻/OT-II naive mice. Tregs were freshly added to the wells at a ratio of 1:4, in the presence of OVA₃₂₃₋₃₃₉ peptide. As a negative control, CAFs were co-cultured with sorted Tregs without OVA₃₂₃₋₃₃₉ peptides. Also, CAFs were co-cultured with Tregs in the presence of OVA₃₂₃₋₃₃₉ peptides and anti-I-A (50 μg/ml, clone M5/114, Bioceros LB). After adding the Treg cells, the slides were placed under a conditioned chamber (37 °C, 5% $CO_2$) of a Leica video microscope for time-lapse imaging. Microphotographs were captured in 10 different representative positions every 5 min for a total of 7–8 h resulting in 10 time-lapse videos per experimental condition.

**Analysis of time-lapse videos.** An in-house-built pipeline was generated to measure contact duration between CAFs and Tregs and to define Treg cell trajectories over time. Treg trajectories throughout the time-lapse videos were detected using the ImageJ plugin Trackmate with the following parameters: estimated object diameter = 8-micron, quality threshold = 1, linking max distance = 36-micron, gap-closing max distance = 12-micron, gap-closing max frame gap = 5. Area of CAFs was determined using the ImageJ plugin Phantast with the following parameters: sigma = 4, epsilon = 0.08. Then, a combination of ImageJ plugins and scripts produced in R environment were used to identify the exact position of CAFs and Tregs for each time-frame in order to determine co-localization between the two cell types and produce the number of time-points that CAF-Treg intersections appear throughout the time-lapse video. All interactions between and CAFs and Tregs observed during a time window of 10 min (2 frames) were quantified and presented in the final output. 4–5 videos corresponding to 7–8 h were analysed in every experimental condition.

**Transmission electron microscopy (TEM).** CAFs were magnetically isolated from fully grown (Day15) melanomas of C57BL/6 mice using the mouse Tumour-Associated Fibroblast Isolation Kit as already described (Miltenyi Biotec). The samples were fixed in a 2.5% glutaraldehyde solution/ 0.1 M sodium cacodylate buffer (SCB) at 4 °C overnight and then were rinsed with SCB (0.1 M, pH 7.2) for 15 min. The rinsing step was repeated for an additional two times. Samples were then fixed with 1% osmium tetroxide for 2 h at RT and rinsed with the 0.1 M SCB buffer three times for 15 min each time. The fixed samples were dehydrated with ethanol solutions with gradient concentrations (30%, 50%, 70%, 80%, 90%, 95%, 100%) for 15 min for each concentration, and treated with 100% dry ethanol once for 20 min, followed by immersing into propylene oxide 2 times for 15 min each time. Samples were then treated with a mixed solution of resin embedding media and propylene oxide(V:V = 1:3) for 1 h, then with a mixed solution of resin embedding media and propylene oxide (V:V = 1:1) for 1 h, then with a mixed solution of resin embedding media and

propylene oxide (V:V = 3:1) for 1 h, and finally with 100% resin embedding media overnight. Afterward, the samples were heated at 60 °C for 48 h. Sections were cut on an Ultramicrotome LEICA EM UC7 at a thickness of 70 nm, then placed on copper 300 mesh grids and stained with lead citrate and uranyl acetate. Electron Microscope JEOL JEM-2100 was used for the observation at 80kv.

**Proteomics.** The cells were lysed in a buffer consisting of 4% SDS, 0.1 M DTT, 100 mM Tris/HCl; pH 7.6 and heated for 5 min at 99 °C. The protein extracts were processed according to the Sp3 protocol. The reduced cysteine residues were alkylated in 200 mM iodoacetamide (Acros Organics). 20 µg of beads (1:1 mixture of hydrophilic and hydrophobic SeraMag carboxylate-modified beads; GE Life Sciences) were added to each sample in 50% ethanol. Protein clean-up was performed on a magnetic rack. The beads were washed twice with 80% ethanol, followed by one wash with 100% acetonitrile (Fisher Chemical). The beads-captured proteins were digested overnight at 37 °C with 0.5 µg trypsin/LysC mix in 25 mM ammonium bicarbonate under vigorous shaking (1200 rpm, Eppendorf Thermomixer). The supernatants were collected and the peptides were purified by a modified Sp3 clean-up protocol and finally solubilized in the mobile phase A (0.1% formic acid in water), and sonicated. Peptide concentration was determined through absorbance measurement at 280 nm using a nanodrop instrument.

Peptides were analysed with liquid chromatography–tandem mass spectrometry (LS–MS/MS) on a setup consisting of a Dionex Ultimate 3000 nanoRSLC online with a Thermo Q Exactive HF-X Orbitrap mass spectrometer. Peptidic samples were directly injected and separated on a 25 cm-long analytical C18 column (PepSep, 1.9 $\mu m^3$ beads, 75 µm ID) using a 90 min long run. The full MS was acquired in profile mode using a Q Exactive HF-X Hybrid Quadropole-Orbitrap mass spectrometer operating in the scan range of 375–1400 $m/z$ using 120 K resolving power with an Automatic Gain Control (AGC) of $3 \times 10^6$ and a max IT of 60 ms followed by data independent analysis (DIA) using 8 Th windows (39 loops counts) with 15 K resolving power with an AGC of $3 \times 10^5$, a max IT of 22 ms and normalized collision energy (NCE) of 26.

Orbitrap raw data were analysed in DIA-NN 1.8 (Data-Independent Acquisition by Neural Networks) against the complete Uniprot Mus musculus proteome (downloaded April 16, 2021). Search parameters were set to allow up to two possible trypsin/P enzyme missed cleavages. A spectra library was generated from the DIA runs and used to re-analyse them. Cysteine carbamidomethylation was set as a fixed modification, while N-terminal acetylation and methionine oxidations were set as variable modifications. The match between runs (MBR) feature was used for all the analyses and the output (precursor) was filtered at 0.01 false discovery rate (FDR). The protein inference was performed on the gene level using only proteotypic peptides. The double pass mode of the neural network classifier was also activated.

## Data analysis and statistics
Data were analysed by GraphPad Prism v8.2.1 software. Comparisons between the two groups were performed using a two-tailed Student's *t*-test or two-tailed Mann–Whitney *U*-test, as appropriate (after testing for normality with the F-test). Multiple group comparisons were performed using one-way analysis of variance (ANOVA) and Dunnett's or Tukey's multiple comparison tests. Kaplan–Meier statistics were evaluated by the Log-rank test. Data are presented as means ± S.E.M. $P$ value < 0.05 was considered as indicative of statistical significance. Compared samples were collected and analysed under the same conditions. G* power analysis was performed (with 90% power and 0.001 type 1 error) to calculate the appropriate sample size. No data were excluded. All data showed normal distribution.

## Reporting summary
Further information on research design is available in the Nature Portfolio Reporting Summary linked to this article.

## Data availability
The data that support the findings of this study are available within the article, Supplementary Information and Source Data file. Source data are provided with this paper; additional data are available from the corresponding author upon reasonable request. The mouse NGS raw data generated in this study are available from the Sequence Read Archive under accession number PRJNA874541. The mass spectrometry proteomics data generated in this study are available from ProteomeXchange Consortium via the PRIDE partner repository under the accession number PXD047024 regarding CAF lysates and PXD047038 regarding CAF secretome. Source data are provided with this paper.

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

## Acknowledgements

The authors thank Anastasia Apostolidou for her technical assistance on flow cytometry and cell sorting, Maria Semitekolou, Katerina Athanasiadou and Christina Papamichail for assisting with handling of human biopsies and assisting with experiments, George Chalepakis for assisting with TEM imaging, Stamatis Pagakis and Eleni Rigana for providing assistance with confocal and time-lapse microscopy, Pavlos Alexakos for assisting with mice, Emmanuel Dialynas and Matthieu Lavigne for assistance with the transcriptomic analysis. A.V. was supported by the "2021 Graduate Student Scholarships" from Fondation Sante and the "3rd Doctorate Scholarships" from the Hellenic Foundation for Research and Innovation (H.F.R.I.) (code: 18529). T.A. was supported by an ERC grant under the European Union's Horizon 2020 research and innovation programme (agreement no. 947975) to T.A., and grants from the Hellenic Foundation for Research and Innovation (H.F.R.I.) under the "2nd Call for H.F.R.I. Research Projects to support Post-Doctoral Researchers" (Project Number: 166) and within the framework of the National Recovery and Resilience Plan Greece 2.0, funded by the European Union—NextGenerationEU (Implementation body: HFRI) (Project Number: 15014) to T.A. This work was also supported by Fondation Sante Research grant to P.V., by the T2EDK-02288, MDS-TARGET from General Secretariat for Research and Technology to P.V. and from the Hellenic Foundation for Research and Innovation (H.F.R.I.) under the action "Always stive for Excellence- Theodoros Papazoglou" grant #1429 to P.V.

## Author contributions

A.V. and M.P. designed and performed experiments, analysed the data, generated the figures. and wrote the manuscript. Z.P. assisted with the mouse RNA-Seq data analysis and scRNA-Seq data meta-analysis and interpretation. E.B.C. performed stainings and microscopy analyses for CAF-Treg synapse visualization. A-I.L. assisted with experiments. M.S. performed proteomic analyses and assisted with the interpretation of proteomics data. A.D. generated the pipe-line for the time-lapse microscopy video analysis. V.D. and P.F. evaluated and provided the human tumour samples. L.B. generated and provided critical reagents. S.P. acquired and interpreted the transmission electron microscopy images. A.H., T.A. and M.L.D. contributed to data interpretation. P.V. designed and supervised the study, performed the data analysis and wrote the manuscript.

## Competing interests

The authors declare no competing interests.
