## [Peer Review File · Nature Communications]

An Immunological Synapse Formation Between T Regulatory Cells and Cancer Associated Fibroblasts Promotes Tumor DevelopmentThis manuscript has been previously reviewed at another journal that is not operating a transparent peer review scheme. This document only contains reviewer comments and rebuttal letters for versions considered at *Nature Communications*.

REVIEWERS' COMMENTS

Reviewer #1 (Remarks to the Author):

The authors have provided additional data to support the presence of an interaction between α SMA+ CAFs and Tregs. Alterations in the secretome of autophagy-deficient α SMA+ CAFs were identified and evidence of reduced antigen presentation are now included. Most of my comments and the comments from Reviewer 2 have been adequately addressed. One minor comment related to indirect effects from other immune cell populations should be addressed, as outlined below.

1. The authors do not analyze the impact of depletion of α SMA+ CAFs on macrophage and dendritic cell populations which could result in reversal of the tolerogenic response. This caveat should be included in the discussion.

Reviewer #2:

[Note from editor: This reviewer was no longer available to comment on the manuscript and was therefore replaced by Reviewer #1].

Reviewer #3 (Remarks to the Author):

In this manuscript by Song N-J. et al., the authors fully answered to the concerns I raised. I have no further concern for publishing this study in Nature Communications.

RESPONSE TO REVIEWERS' COMMENTS

Reviewer #1 (Remarks to the Author):

The authors have provided additional data to support the presence of an interaction between α SMA+ CAFs and Tregs. Alterations in the secretome of autophagy-deficient α SMA+ CAFs were identified and evidence of reduced antigen presentation are now included. Most of my comments and the comments from Reviewer 2 have been adequately addressed. One minor comment related to indirect effects from other immune cell populations should be addressed, as outlined below.

1. The authors do not analyze the impact of depletion of α SMA+ CAFs on macrophage and dendritic cell populations which could result in reversal of the tolerogenic response. This caveat should be included in the discussion.

We thank the Reviewer for the comment. As proposed, we comment further on the impact of α -SMA+ CAF depletion on myeloid derived cell populations (discussion part, page 16 of revised manuscript).

Reviewer #2:

[Note from editor: This reviewer was no longer available to comment on the manuscript and was therefore replaced by Reviewer #1].

Reviewer #3 (Remarks to the Author):

In this manuscript by Varveri et al., the authors fully answered to the concerns I raised. I have no further concern for publishing this study in Nature Communications.